# Sexual dimorphism: increased sterol excretion leads to hypocholesterolaemia in female hyperbilirubinaemic Gunn rats

Josif Vidimce[1], Johara Pillay[1], Onne Ronda[2], Ai-Ching Boon[1], Evan Pennell[1], Kevin J. Ashton[3] ⓘD, Theo H. van Dijk[4] ⓘD, Karl-Heinz Wagner[5], Henkjan J. Verkade[2] and Andrew C. Bulmer[1] ⓘD

[1] *School of Pharmacy and Medical Sciences, Griffith University, Gold Coast, Queensland, Australia*
[2] *Pediatric Gastroenterology/Hepatology, Department of Pediatrics, University of Groningen, University Medical Center Groningen, Groningen, The Netherlands*
[3] *Faculty of Health Science and Medicine, Bond University, Gold Coast, Australia*
[4] *Department of Laboratory Medicine, University of Groningen, University Medical Center Groningen, Groningen, The Netherlands*
[5] *Department of Nutritional Sciences and Research Platform Active Ageing, University of Vienna, Vienna, Austria*

Edited by: Kim Barrett & Kyle McCommis

The peer review history is available in the supporting information section of this article (https://doi.org/10.1113/JP282395#support-information-section).

K-H. Wagner and H. J. Verkade contributed equally to this work.

The Journal of Physiology

**Abstract**   Circulating bilirubin is associated with reduced serum cholesterol concentrations in humans and in hyperbilirubinaemic Gunn rats. However, mechanisms contributing to hypocholesterolaemia remain unknown. Therefore, this study aimed to investigate cholesterol synthesis, transport and excretion in mutant Gunn rats. Adult Gunn and control rats were assessed for daily faecal sterol excretion using metabolic cages, and water was supplemented with $[1-^{13}C]$-acetate to determine cholesterol synthesis. Bile was collected to measure biliary lipid secretion. Serum and liver were collected for biochemical analysis and for gene/protein expression using RT-qPCR and western blot, respectively. Additionally, serum was collected and analysed from juvenile rats. A significant interaction of sex, age and phenotype on circulating lipids was found with adult female Gunn rats reporting significantly lower cholesterol and phospholipids. Female Gunn rats also demonstrated elevated cholesterol synthesis, greater biliary lipid secretion and increased total faecal cholesterol and bile acid excretion. Furthermore, they possessed increased hepatic low-density lipoprotein (LDL) receptor and SREBP2 expression. In contrast, there were no changes to sterol metabolism in adult male Gunn rats. This is the first study to demonstrate elevated faecal sterol excretion in female hyperbilirubinaemic Gunn rats. Increased sterol excretion creates a negative intestinal sterol balance that is compensated for by increased cholesterol synthesis and LDL receptor expression. Therefore, reduced circulating cholesterol is potentially caused by increased hepatic uptake via the LDL receptor. Future studies are required to further evaluate the sexual dimorphism of this response and whether similar findings occur in females with benign unconjugated hyperbilirubinaemia (Gilbert's syndrome).

(Received 20 September 2021; accepted after revision 2 February 2022; first published online 14 February 2022)
**Corresponding author** A. C. Bulmer: School of Pharmacy and Medical Sciences, Griffith University, Gold Coast, Queensland, Australia.      Email: a.bulmer@griffith.edu.au

**Abstract figure legend** Female Gunn rats have an unconjugated hyperbilirubinaemia because of dysfunctional UGT1A1 enzyme function. Through an unknown mechanism, UGT1A1 dysfunction and/or elevated unconjugated bilirubin concentrations lead to increased faecal sterol excretion. Increased sterol outflow is potentially compensated by increased SREBP2-mediated cholesterol synthesis and LDL receptor expression, ultimately contributing to reduced circulating cholesterol concentrations.

## Key points

- Female adult hyperbilirubinaemic (Gunn) rats demonstrated lower circulating cholesterol, corroborating human studies that report a negative association between bilirubin and cholesterol concentrations.
- Furthermore, female Gunn rats had elevated sterol excretion creating a negative intestinal sterol balance that was compensated for by elevated cholesterol synthesis and increased hepatic low-density lipoprotein (LDL) receptor expression.
- Therefore, elevated LDL receptor expression potentially leads to reduced circulating cholesterol levels in female Gunn rats providing an explanation for the hypocholesterolaemia observed in humans with elevated bilirubin levels.
- This study also reports a novel interaction of sex with the hyperbilirubinaemic phenotype on sterol metabolism because changes were only reported in females and not in male Gunn rats.
- Future studies are required to further evaluate the sexual dimorphism of this response and whether similar findings occur in females with benign unconjugated hyperbilirubinaemia (Gilbert's syndrome).

## Introduction

Cardiovascular diseases (CVDs) cause the greatest number of deaths globally, with atherosclerosis being an important contributor. The development of atherosclerosis is dependent on the complex interplay between rising plasma cholesterol concentrations and systemic inflammation. Thus, elevated cholesterol concentrations are a critical risk factor for CVD mortality (Tabas *et al.* 2015). Despite the effectiveness of statins to reduce plasma cholesterol concentrations, they only reduce the clinical manifestations of atherosclerosis by 20–40% and carry an important side effect profile limiting their long-term use in some patient groups (Insull, 2009). Therefore, there is an important need for more effective medication to prevent and treat CVDs.

Although formerly considered a toxic breakdown product of haem catabolism, growing evidence supports a protective role for unconjugated bilirubin (UCB) against CVDs (Horsfall *et al.* 2013; Bulmer *et al.* 2018). Furthermore, individuals with Gilbert's syndrome (GS) who possess mildly elevated plasma UCB concentrations due to impaired UGT1A1 function are protected from coronary artery disease, ischaemic heart disease, atherosclerosis and all-cause mortality (Hopkins *et al.* 1996; Novotný & Vítek, 2003; Lin *et al.* 2006; Horsfall *et al.* 2013). Initially, UCB was demonstrated to possess potent antioxidant activity against peroxyl radicals and against low-density lipoprotein (LDL) oxidation, providing one potential mechanism to explain CVD protection (Stocker *et al.* 1987; Neuzil & Stocker, 1994; Bulmer *et al.* 2008; Horsfall *et al.* 2013), although additional mechanisms likely contribute (Bulmer *et al.* 2018).

Interestingly, an observational study reported that GS individuals demonstrate reduced serum concentrations of oxidized LDL (oxLDL) *and* total LDL cholesterol (LDL-C) (Boon *et al.* 2012). Therefore, it is unclear whether the antioxidant properties of UCB reduce the formation of oxLDL or whether reduced oxLDL concentrations are secondary to the overall reduction in LDL-C found with increasing plasma UCB concentrations. Recent epidemiological studies report that UCB is associated with lower total cholesterol and LDL-C, indicating that UCB potentially also regulates cholesterol metabolism in addition to functioning as an antioxidant (Bulmer *et al.* 2013; Seyed Khoei *et al.* 2018). In agreement, female hyperbilirubinaemic (Gunn) rats also report >50% lower serum cholesterol concentrations than normobilirubinaemic controls. However, in contrast to human studies, the hypocholesterolaemic effects appear to be sex-specific in rats with a smaller reduction observed in male Gunn rats (Wallner *et al.* 2013). Similarly, we have observed profound sexual dimorphism in body composition, mitochondrial function and fat metabolism in hyperbilirubinaemic Gunn rats, the reason for which remains unknown (Vidimce *et al.* 2021). However, these studies suggest that UCB and/or UGT1A1 impairment may reduce circulating cholesterol concentrations, distinct from UCB's antioxidant properties (Bulmer *et al.* 2013).

Recent studies demonstrate that UCB is a peroxisome proliferator-activated receptor alpha (PPARα) agonist, similar in potency to fenofibrate (Stec *et al.* 2016). Additionally, UCB is an atypical agonist of the Aryl hydrocarbon receptor (AhR) (Phelan *et al.* 1998). PPARα and AhR are ligand-activated transcription factors that regulate genes involved in lipid metabolism, and their activation may affect circulating cholesterol and triglyceride concentrations (Yoon, 2009; Tanos *et al.* 2012). The capacity of UCB to activate these receptors may help to explain the hypocholesterolaemic phenotype observed in GS and hyperbilirubinaemic animals; however, the role of these pathways in hyperbilirubinaemia remains poorly understood (Bulmer *et al.* 2013). Taken together, there is mounting evidence that UCB and/or UGT1A1 impairment can reduce serum cholesterol concentrations. However, nothing is known about whether these effects are mediated by decreased *de novo* synthesis or increased cholesterol clearance (Bulmer *et al.* 2013). Therefore, the aim of this study was to determine which aspects of sterol metabolism including synthesis, clearance and hepatic transport, lead to hypocholesterolaemia in Gunn rats and to evaluate whether these effects are sex-dependent.

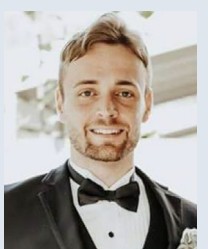

**Josif Vidimce** recently completed his PhD at the School of Medical Sciences at Griffith University, Australia, and is now studying clinical medicine, hoping to become a clinical researcher in the future. His current research focuses on the impact of bilirubin on metabolism and whether mild increases in this endogenous molecule can protect against metabolic diseases. Josif's research interests also expand into mitochondrial physiology and lipid metabolism. He has published 10 peer-reviewed journal articles and has presented his research at various national and international conferences.

**Table 1. Schematic detailing the timing of various procedures and assessed parameters**

| Procedures | −1 | 0 | 1 | 2 | 3 | 4 | 5 | 6 | 7 | 8 | 9 | 10 | 11 | 12 | 13 | 14 | 15 | 16 | 17 | 18 | 19 |
|---|---|---|---|---|---|---|---|---|---|---|---|---|---|---|---|---|---|---|---|---|---|
| Body weight (all groups) | ↑ | ↑ | ↑ |  | ↑ |  | ↑ |  | ↑ |  | ↑ |  | ↑ |  | ↑ |  | ↑ |  | ↑ |  | ↑ |
| Metabolic cage (food intake, faeces/urine collection) | ↑ |  |  |  |  |  |  |  |  |  |  | ↑ |  |  |  | ↑ |  |  |  | ↑ |  |
| Blood samples (above)/blood spots (below) |  | ↑ |  |  |  |  |  |  | ↑ |  |  |  |  |  |  |  | ↑ | ↑↑ | ↑↑ | ↑↑ | ↑ |
| Vehicle I.P. injections |  |  | ↑ |  | ↑ |  | ↑ |  | ↑ |  | ↑ |  | ↑ |  | ↑ |  | ↑ |  | ↑ |  | ↑ |
| $H_2O$ oral gavage |  |  | ↑ | ↑ | ↑ | ↑ | ↑ | ↑ | ↑ | ↑ | ↑ | ↑ | ↑ | ↑ | ↑ | ↑ | ↑ | ↑ | ↑ | ↑ | ↑ |
| [$^{13}$C]-acetate |  |  |  |  |  |  |  |  |  |  |  |  |  |  |  |  |  | ↑ | ↑ | ↑ | ↑ |
| Terminal procedures and bile duct cannulation |  |  |  |  |  |  |  |  |  |  |  |  |  |  |  |  |  |  |  |  | ↑ |

Note: arrow (↑) indicates on which day a selected procedure is performed. Double arrow (↑↑) indicates that the procedure was performed twice (12 h apart).

## Methods

### Ethics approval

All procedures were approved by the Griffith University Animal Ethics Research Committee (MSC/02/17/AEC) prior to experimentation. All experiments were carried out according to the Griffith University animal welfare policy and to the principles and regulations as described in the Editorial by Grundy (Grundy, 2015).

### Chemicals and reagents

All chemicals and reagents were purchased from Sigma Aldrich (Australia) unless otherwise stated.

### Animals

Breeding pairs of heterozygote Gunn rats on a Wistar background were imported from the Rat Research and Resource Centre (Columbia, MO, USA) and cross-bred to produce both homozygote (hyperbilirubinaemic) and heterozygote (normobilirubinaemic) Gunn rats. Gunn rats were determined phenotypically to be homozygote based on the presence of jaundice in the days after birth and were ear-tagged. The hyperbilirubinaemic phenotype was then confirmed by measuring serum bilirubin concentrations. This Gunn colony is regularly backcrossed with female wild-type Wistar rats, supplied by the Animal Resources Centre (Canning Vale, WA, Australia). From this point, animals expressing hyperbilirubinaemia are referred to as 'Gunn' rats while littermates with normal bilirubin levels are indicated as 'controls'. Rats were housed in an approved animal house facility at Griffith University, Gold Coast, at constant temperature (20°C) and humidity (60%), with a 12 h light:dark cycle. All animals were provided a standard rodent diet (18% protein, 6.2% fat, cholesterol-free; TEKLAD Standard Global, USA) and water *ad libitum*.

### Experimental protocol

**Adult rats.** Thirty six age-matched adult rats (∼10 weeks old) were separated into four groups based on sex and phenotype (control female ($n = 9$), Gunn female ($n = 10$), control male ($n = 8$) and Gunn male ($n = 9$)). Rats were gradually acclimatized to metabolic cages for 2, 5 and then 24 h before entering a 21-day protocol (days -1 to 19; Table 1). On days -1, 10, 14 and 18, animals were housed individually in metabolic cages for a period of 24 h at a time. Food and water were freely available and were weighed before and after each animal was housed in metabolic cages. Additionally, urine and faeces were collected, weighed and stored after each metabolic cage day, and the averaged values of the four individual metabolic cage days was used to report daily food and water consumption, and faecal and urine outputs. Faeces were air-dried for at least 48 h prior to weighing. Body weight was recorded every 2 days for the duration of the study and fasted (6–7 h) blood samples (approx. 1 ml) were collected at day 0 and day 7 via tail bleed (see *Blood collection*). During the final four days of the study, drinking water was supplemented with [1-$^{13}$C]-acetate (2% w/v) and blood spots (from the tail) were collected twice daily from non-fasted animals (see *Blood collection*) to determine *in vivo* cholesterol synthesis. On the final day of the study, animals underwent terminal day procedures (see *Bile duct cannulation and terminal procedures*) and

were euthanized via surgical removal of the heart under anaesthesia. To control for interventions within future studies, animals were administered 400 $\mu$l of drinking water daily via oral gavage and 800 $\mu$l of sterile phosphate buffered saline (Gibco, United Kingdom) every 2 days via intraperitoneal injection (I.P.) for the duration of the study.

**Juvenile rats.** Juvenile rats (3–4 weeks of age) were separated based on sex and phenotype (control females ($n = 5$), Gunn females ($n = 7$), control males ($n = 8$), Gunn males ($n = 10$)). All animals were fasted for 6 h prior to tissue collection and were anaesthetized using 50 mg kg$^{-1}$ sodium pentobarbitone (Lethabarb, Australia) via an I.P. injection. Pedal reflexes were checked to ensure the animals were non-responsive to pain. Blood was then collected via the inferior vena cava and animals were euthanized by surgical removal of the heart. Blood was centrifuged (2000 $g$, 10 min, 21°C) and the serum was flash-frozen in liquid N$_2$ and stored at -80°C for later analysis.

### Blood collection

Animals were anaesthetized using 2–5% isofluorane (Lyppard, Australia) in 100% oxygen via inhalation (1–2 l min$^{-1}$). Pedal reflexes were routinely checked, and the tip of the tail (1–2 mm) was removed using a sterile scalpel blade to draw blood. The first drop of blood was discarded before collecting ~1 ml of blood in a 1.5 ml microtube (Sarstedt, Australia) or a ~20 $\mu$l blood spot (~6 mm in diameter) on a filter paper (Sartorius, France). Blood was centrifuged (2000 $g$, 10 mins at 21°C) and the serum was flash-frozen and stored at -80°C for later analysis. Blood spots were left to air-dry for at least 24 h prior to storage at room temperature (RT).

### Bile duct cannulation and terminal procedures

On day 19 of the protocol, rats were anaesthetized using a ketamine (50 mg kg$^{-1}$) and xylazine (3 mg kg$^{-1}$) mixture via I.P. injection and additional anaesthetic was administered as necessary to maintain a surgical plane. Pedal reflexes were routinely checked prior to and during the procedure. Body temperature was maintained using a heating pad and the temperature was monitored with a rectal probe. A midline laparotomy allowed access to and identification of the common bile duct, with the aid of a dissecting microscope. The bile duct was cannulated with 0.78 o.d. × 0.32 i.d. mm tubing (Micro-tube Extrusions, Australia) and secured in place with a drop of superglue and sutures. Once the bile duct was secured, the abdomen was closed and secured with sutures to maintain body temperature. Bile was collected for 35 min, with the first 5 min discarded while the rest

was collected into pre-weighed 1.5 ml microtubes on ice. During the procedure, bile was protected from light by covering the collection tube with foil. Immediately after bile collection, the tubes were weighed, flash-frozen in liquid N$_2$ and stored at -80°C. Bile flow was determined gravimetrically assuming a density of 1 g ml$^{-1}$. Rates of lipid secretion were calculated using bile flow and biliary lipid concentrations. After surgery, blood was collected from the inferior vena cava and centrifuged (2000 $g$, 10 min, 4°C). The supernatant was flash-frozen in liquid N$_2$ and stored at -80°C. Next, the rats were euthanized by surgical removal of the heart. Liver tissue was rinsed with cold dPBS (Gibco, United Kingdom) and one section was flash-frozen with liquid N$_2$ and stored at -80°C while the other was stored in RNAlater solution (Invitrogen, Australia) at 4°C for at least 24 h prior to storage at -80°C.

### Biochemistry analysis

Analysis of serum, bile, liver and faecal biochemistry were conducted using enzymatic, colorimetric assays on the COBAS Integra 400+ (Roche Diagnostics, Australia) or using gas chromatography mass spectrometry (GS/MS) (see *Gas chromatography mass spectrometry analysis*). Lipid concentrations were assessed using total cholesterol (CHOL Gen. 2, Roche Diagnostics), high-density lipoprotein cholesterol (HDL-C Gen. 3, Roche Diagnostics), triglycerides (TRIGL, Roche Diagnostics), total bile acids (Total Bile Acids #BI3863, RANDOX, Australia) and phospholipids (Phospholipids C #997-01801, Wako, Japan). Additionally, samples were analysed for total bilirubin (BIL-T Gen. 3, Roche Diagnostics) and albumin (ALB Gen. 2, Roche Diagnostics). All biochemical analysis kits were verified with their appropriate calibrators and quality controls prior to sample analysis.

### Gas chromatography mass spectrometry (GC/MS) analysis

Cholesterol label enrichment and measurement of fractional *de novo* cholesterol synthesis rate, biliary cholesterol concentrations and biliary bile acid species were measured as previously published (Ronda *et al.* 2016). Briefly, cholesterol was extracted from blood spots or bile and derivatized with N,O-Bis (trimethylsilyl) trifluoroacetamide (BSTFA) containing 1% trimethylchlorosilane before analysis with GS/MS (Agilent 7890A and 5975c, respectively; Netherlands). Isotope ratios were determined by selected ion monitoring mode on m/z 458 (M0) to 465 (M7). The fractional rate of cholesterol synthesis was calculated according to mass isotopomer distribution analysis (Ronda *et al.* 2016). Biliary bile acids were extracted and derivatized with

BSTFA-pyridine-trimethylchlorosilane (5:5:0.1) and were quantified using GC/MS. Samples that contained biliary bile acid species below the quantification limit were calculated by dividing the limit of quantification by the square root of two as previously published (Glass & Grey, 2001). The hydrophobicity index of biliary bile acid composition was calculated according to Heuman *et al.* (1989).

### Liver and faecal lipid extraction

Frozen liver tissue (∼100 mg) and dried faeces were homogenized using a mortar and pestle and stored at -80°C or RT, respectively, for later lipid extraction. Cholesterol was extracted from homogenized liver and faecal samples by three sequential solvent extractions with isopropyl-alcohol at a 50:1 mg:ml ratio of sample (liver or faeces) to isopropyl-alcohol as previously published (Vidimce *et al.* 2021). During each extraction, homogenates were vortexed vigorously and sonicated at RT for 10 min prior to centrifugation (43,000 *g*, 20°C, 10 min). The three supernatant fractions were collected in a separate tube and evaporated using a rotary evaporator (Maxivac, Labogene, Australia) at 35°C. Dry pellets were reconstituted in 250 $\mu$l of isopropanol and analysed for total cholesterol (see *Biochemistry analysis*). Bile acids were extracted from homogenized faecal samples using a solvent mixture of 1 M sodium hydroxide and methanol (1:3). Briefly, 1 ml of solvent mixture was added to each 40 mg of homogenized faecal sample and then the sample mixtures were heated for 2 h at 80°C. The samples were then allowed to reach RT before being diluted in $H_2O$ and centrifuged (43,000 *g*, 20°C, 10 min). The supernatants were analysed for total bile acid concentration (see *Biochemistry analysis*).

### RT-qPCR array

Liver samples preserved in RNAlater were homogenized in TRIzol reagent (Invitrogen, Australia), and the total RNA was isolated with RNA extraction kits (Qiagen, Australia) as previously published (Ashton *et al.* 2013). The concentration and purity of RNA was analysed using spectrophotometry (NanoDrop, ThermoFisher Scientific, Australia). First-strand cDNA synthesis kits (Qiagen) were used to synthesize cDNA from total RNA (500 ng) according to the manufacturer's protocol and synthesized cDNA was stored at -20°C for later analysis. Rat Lipoprotein Signalling and Cholesterol Metabolism RT$^2$ Profiler PCR Arrays (#PARN-080Z, Qiagen) were utilized to measure the expression of 84 key genes using the CFX96 Real-Time PCR Detection System (Bio-Rad, Australia). All data acquisition was performed with CFX Manager 2.0 software (Bio-Rad). Relative mRNA expression was calculated using the $2\hat{ }(-\Delta\Delta CT)$ method to calculate the relative changes of Gunn *versus* control, with normalization of CT values performed against *Rplp1* (ribosomal protein lateral stalk subunit P1) as the most stable reference gene.

### Western blot

Frozen liver samples were homogenized by shearing with 18–23G needles in CelLytic MT Cell Lysis buffer (#C3228) in the presence of protease inhibitor (#P8340) and phosphatase inhibitor (#ab201114, Abcam, USA) as per the manufacturer's protocol. The tissue supernatant was generated by centrifugation (4000 *g* for 10 min, 4°C) and standardized based on protein concentration using the Pierce BCA Protein Assay Kit (ThermoFisher Scientific, Australia) as per the manufacturer's protocol. Standardized samples were prepared in Laemmli 2X buffer (#S3401) at a 1:1 ratio and were heated for 5 min at 95°C or unheated (ABCA1 protein), prior to loading. Proteins (10–30 $\mu$g) were separated on SDS-PAGE (7.5, 10 or 12%) using TGX FastCast acrylamide kits (#1610171, #1610173 and #1610175, Bio-Rad). After electrophoresis, proteins were transferred onto 0.45 $\mu$m polyvinylidene fluoride membranes (#IPFL0010, Millipore, Australia) for 1–2 h on ice, in Towbin buffer (25 mM Tris, 192 mM glycine and 20% v/v methanol). Following transfer, membranes were blocked with Odyssey Blocking Buffer (Millenium Science, Australia) and incubated with primary antibodies (HMGCR, #ab174830; SREBP2, #ab30682; B-actin, #ab8226; GAPDH, #sc-32233; CYB5R3, #ab109620; LDLr, #ab30532; CYP7A1, #ab65596; ABCA1, #ab18180) overnight at 4°C with gentle agitation. Finally, membranes were incubated with secondary antibodies (#IRDye 680 or #IRDye 800CW, LI-COR, Australia) for 1 h with gentle agitation at RT and visualized using the Odyssey CLX Infrared Imaging System (LI-COR). Densitometric analysis of target proteins were normalized to GAPDH or $\beta$-actin. The linear range of detection was determined for each target and housekeeping (GAPDH and $\beta$-actin) proteins prior to analysis. All densitometric analyses were conducted using 5.2 Image Studio Lite (LI-COR).

### Statistical analysis

All values are expressed as means $\pm$ SD or median and interquartile range (IQR). Data were tested for normality and homogeneity of variance with the Kolmogorov–Smirnov and Spearman tests, respectively. Two-way or three-way ANOVA was used to compare the main effects of phenotype (Gunn *vs.* control), sex (male *vs.* female) and age (juvenile vs adult) between groups unless otherwise stated. *Post hoc* comparisons were conducted between phenotype (Gunn *vs.* control

rats) of the same age and within the same sex using Fisher's least significant difference test. Statistical analysis was performed in GraphPad PRISM (v9.0) and a $P < 0.05$ was considered statistically significant.

## Results

### Perturbed serum lipid concentrations in adult female Gunn rats

A total of 36 adult and 30 juvenile animals (Gunn and littermate controls) were assessed for circulating lipid concentrations (see Table 2). Hyperbilirubinaemia was confirmed in adult and juvenile Gunn rats (Main effect: phenotype: $P < 0.0001$; Table 2). Total cholesterol and HDL-C were significantly reduced in Gunn rats (phenotype: $P < 0.0001$; Table 2). A significant interaction between phenotype, sex and age (total cholesterol interaction: $P = 0.0011$; HDL-C interaction: $P < 0.0001$; Table 2) was reported with *post hoc* analysis revealing that total cholesterol and HDL-C were significantly reduced in adult, but not in juvenile, female Gunn rats (*post hoc*: $P < 0.0001$; Table 2). The opposite was observed in male rats, with a significant reduction in total cholesterol (*post hoc*: $P = 0.0002$) and HDL-C (*post hoc*: $P = 0.0060$) in juvenile, but not in adult, Gunn rats (Table 2). Phospholipid concentrations were significantly reduced in Gunn rats (phenotype: $P = 0.0003$; Table 2). *Post hoc* analysis demonstrated that phospholipids were significantly lower in male juvenile (*post hoc*: $P = 0.0121$) and female adult Gunn (*post hoc*: $P = 0.0121$) rats compared with controls of the same age and sex (Table 2).

### Adult female Gunn rats have reduced body weight and consume fewer calories

Adult rats were housed in metabolic cages for 24 h on four different days to assess daily food, water intake and faecal/urine output (Table 3). Gunn rats had lower body weight than controls (phenotype: $P < 0.0001$; male Gunn *vs.* controls *post hoc*: $P = 0.0129$; female Gunn *vs.* control *post hoc*: $P < 0.0001$; Table 3). *Post hoc* analysis revealed a significantly lower food intake in female Gunn compared with control rats (*post hoc*: $P = 0.0147$; Table 3).

### Adult female Gunn rats demonstrate elevated fractional cholesterol synthesis

To investigate cholesterol synthesis *in vivo*, acetate pools were enriched with [1-$^{13}$C]-acetate supplementation in drinking water. Fractional cholesterol synthesis reflects the *in vivo* rate of cholesterol synthesis determined by the fractional contribution of newly synthesized [$^{13}$C]-enriched cholesterol ([$^{13}$C]-cholesterol) to the over-

all circulating pool of cholesterol at steady state. There was a significant interaction between phenotype and sex on the rate of fractional cholesterol synthesis (interaction: $P = 0.0170$; Fig. 1$A$). *Post hoc* analysis demonstrated that female Gunn rats had a significantly greater rate of serum fractional cholesterol synthesis ($33.8 \pm 3.77$ *vs.* $28.4 \pm 5.73$%; *post hoc*: $P = 0.0272$), while male Gunn *vs.* control rats showed no difference ($18.1 \pm 5.98$% *vs.* $21.3 \pm 4.40$%; *post hoc*: $P = 0.210$; Fig. 1$A$). Considering that circulating cholesterol levels were lower despite an increased rate of cholesterol synthesis in female Gunn rats, hepatic cholesterol content was assessed. However, no difference in hepatic cholesterol content existed (phenotype: $P = 0.901$; interaction: $P = 0.961$; Fig. 1$B$).

### Increased biliary and faecal sterol output in adult female Gunn rats

Since hepatic cholesterol content was not different between Gunn and control female rats, it was hypothesized that sterol excretion was enhanced in female Gunn rats, creating a negative sterol balance that is compensated for by increased cholesterol synthesis. Therefore, biliary secretion and faecal lipid excretion were assessed. Bile flow (phenotype: $P = 0.0092$), biliary cholesterol (phenotype: $P = 0.0008$) and phospholipid (phenotype: $P = 0.0009$) secretion were significantly greater in Gunn rats (Fig. 2$A$–$C$). Furthermore, there was a significant interaction between sex and phenotype on biliary cholesterol (interaction: $P = 0.0411$) and phospholipid (interaction: $P = 0.0182$) secretion (Fig. 2). *Post hoc* analysis demonstrated that female Gunn rats had significantly greater bile flow ($0.67 \pm 0.13$ *vs.* $0.47 \pm 0.06$ ml h$^{-1}$ 100 g$^{-1}$ body weight; *post hoc*: $P = 0.0035$), biliary cholesterol ($232 \pm 32.7$ *vs.* $141 \pm 42.1$ nmol h$^{-1}$ 100 g$^{-1}$ body weight; *post hoc*: $P = 0.0004$) and biliary phospholipid secretion ($1.69 \pm 0.30$ *vs.* $0.97 \pm 0.36$ mg h$^{-1}$ 100 g$^{-1}$ body weight; *post hoc*: $P = 0.0003$; Fig. 2$A$–$C$) while there was no difference between male Gunn rats and controls. Additionally, female Gunn rats had greater biliary bile acid secretion ($7.34 \pm 1.75$ *vs.* $5.50 \pm 1.67$ $\mu$mol h$^{-1}$ 100 g$^{-1}$ body weight; *post hoc*: $P = 0.0256$; Fig. 2). Biliary lipid (cholesterol + phospholipid) secretion relative to bile acid output was elevated in Gunn rats (phenotype: $P = 0.0018$; $0.33 \pm 0.06$ *vs.* $0.24 \pm 0.03$ mol:mol, lipid:bile acids, female Gunn *vs.* controls; *post hoc*: $P = 0.0073$; $0.31 \pm 0.07$ *vs.* $0.25 \pm 0.04$ mol:mol, lipid:bile acids, male Gunn *vs.* controls; *post hoc*: $P = 0.0476$; Fig. 2$E$).

Daily faecal cholesterol (day 14 phenotype: $P = 0.0449$; day 18 phenotype: $P = 0.0144$; Fig. 3$A$–$B$), bile acids (day 14 phenotype: $P < 0.0001$; day 18 phenotype: $P = 0.0002$; Fig. 3$C$–$D$) and total sterol (day 14 phenotype: $P < 0.0001$; day 18 phenotype: $P = 0.0003$; Fig. 3$E$–$F$) excretion

**Table 2. Serum biochemistry of juvenile and adult rats**

| Variable | Juvenile† Control | Juvenile† Gunn | Adult‡ Control | Adult‡ Gunn | Phenotype[a] | Sex[b] | Age[c] | Three-way ANOVA Interaction | Post hoc |
|---|---|---|---|---|---|---|---|---|---|
| **Albumin (g l⁻¹)** | | | | | | | | | |
| Males | 30.7 (3.25) | 27.1 (7.66) | 40.2 (2.87) | 43.4 (3.30) | 0.559 | **0.0227** | **< 0.0001** | $0.492^{a*b}$, $0.672^{a*c}$ | 0.154†, 0.211‡ |
| Females | 30.5 (1.72) | 34.5 (8.86) | 44.7 (5.08) | 44.1 (3.32) | | | | $0.732^{b*c}$, $\mathbf{0.0352^{a*b*c}}$ | 0.194†, 0.809‡ |
| **Total bilirubin ($\mu$mol l⁻¹)** | | | | | | | | | |
| Males | 2.29 (1.53) | 72.6 (32.0) | 2.29 (1.02) | 109 (15.0) | **<0.0001** | 0.0865 | 0.377 | $0.0896^{a*b}$, $0.369^{a*c}$ | **<0.0001†, <0.0001‡** |
| Females | 2.28 (0.54) | 83.9 (34.2) | 2.13 (1.54) | 64.8 (13.8) | | | | $\mathbf{0.0047^{b*c}}$, $\mathbf{0.0049^{a*b*c}}$ | **<0.0001†, <0.0001‡** |
| **Total cholesterol (mmol l⁻¹)** | | | | | | | | | |
| Males | 1.79 (0.51) | 1.21 (0.30) | 1.56 (0.23) | 1.41 (0.15) | **<0.0001** | **0.0005** | 0.119 | $0.129^{a*b}$, $0.439^{a*c}$ | **0.0002†**, 0.333‡ |
| Females | 1.45 (0.32) | 1.17 (0.34) | 1.56 (0.34) | 0.60 (0.12) | | | | $0.185^{b*c}$, $\mathbf{0.0011^{a*b*c}}$ | 0.128†, **<0.0001‡** |
| **HDL–C (mmol l⁻¹)** | | | | | | | | | |
| Males | 1.23 (0.43) | 0.89 (0.25) | 1.33 (0.22) | 1.37 (0.16) | **<0.0001** | **<0.0001** | 0.191 | $\mathbf{0.0002^{a*b}}$, $\mathbf{0.0171^{a*c}}$ | **0.0060†**, 0.772‡ |
| Females | 0.99 (0.24) | 0.82 (0.32) | 1.39 (0.25) | 0.20 (0.09) | | | | $\mathbf{0.0033^{b*c}}$, $\mathbf{<0.0001^{a*b*c}}$ | 0.272†, **<0.0001‡** |
| **Triglycerides (mmol l⁻¹)** | | | | | | | | | |
| Males | 0.93 (0.27) | 0.88 (0.32) | 1.04 (0.56) | 1.19 (0.52) | 0.447 | 0.580 | 0.457 | $0.747^{a*b}$, $0.721^{a*c}$ | 0.811†, 0.508‡ |
| Females | 0.89 (0.08) | 1.04 (0.55) | 0.87 (0.66) | 0.98 (0.37) | | | | $0.309^{b*c}$, $0.605^{a*b*c}$ | 0.579†, 0.633‡ |
| **Phospholipids (mg dl⁻¹)** | | | | | | | | | |
| Males | 156 (38.0) | 120 (21.6) | 128 (18.1) | 129 (12.9) | **0.0003** | **0.0382** | **0.0432** | $0.290^{a*b}$, $0.310^{a*c}$ | **0.0023†**, 0.953‡ |
| Females | 141 (17.9) | 115 (26.7) | 130 (29.2) | 94.0 (11.9) | | | | $0.648^{b*c}$, $0.0604^{a*b*c}$ | 0.0725†, **0.0034‡** |
| **Total bile acids ($\mu$mol l⁻¹)** | | | | | | | | | |
| Males | 100 (72.7) | 93.2 (61.3) | 16.0 (11.0) | 14.2 (4.47) | 0.907 | **0.0205** | **<0.0001** | $0.822^{a*b}$, $0.317^{a*c}$ | 0.766†, 0.947‡ |
| Females | 157 (86.0) | 134 (69.3) | 16.3 (7.35) | 41.7 (27.1) | | | | $0.190^{b*c}$, $0.421^{a*b*c}$ | 0.449†, 0.278‡ |

Note: Control group represents normobilirubinaemic heterozygote littermates. Gunn group represents hyperbilirubinaemic homozygote littermates. Juveniles were 3–4 weeks of age (females: control $n = 5$, Gunn $n = 7$; males: control $n = 8$, Gunn $n = 10$). Adult rats were 14 weeks of age (females: control $n = 9$, Gunn $n = 10$; males: control $n = 8$, Gunn $n = 9$). Abbreviations: HDL-C, high-density lipoprotein cholesterol. Values are represented as means (SD). Three-way ANOVA was performed with main effects: phenotype (Gunn or control), sex (male or female), and age (juvenile or adult). All *post hoc* analyses compared differences between phenotype of the same age group within the same sex († = juvenile; ‡ = adult). Statistically significant ($P < 0.05$) $P$ values are highlighted in bold.

**Table 3. Terminal body weight, daily food and water intake, and excretion of urine and faeces in adult rats**

| | Phenotype | | Two-way ANOVA | | | |
|---|---|---|---|---|---|---|
| Variable | Control | Gunn | Phenotype | Sex | Interaction | *Post hoc* |
| **Body weight (g)** | | | | | | |
| Males | 416 (42.7) | 382 (27.4) | **<0.0001** | **<0.0001** | 0.246 | **0.0122** |
| Females | 246 (16.2) | 191 (12.8) | | | | **<0.0001** |
| **Food intake (kcal day$^{-1}$)** | | | | | | |
| Males | 80.6 (13.1) | 79.2 (6.12) | 0.0501 | **<0.0001** | 0.141 | 0.720 |
| Females | 61.9 (5.72) | 52.5 (5.25) | | | | **0.0147** |
| **Water intake (g day$^{-1}$)** | | | | | | |
| Males | 33.3 (8.95) | 32.5 (7.13) | 0.423 | 0.316 | 0.584 | 0.860 |
| Females | 32.0 (9.04) | 28.3 (7.41) | | | | 0.329 |
| **Urine output (g day$^{-1}$)** | | | | | | |
| Males | 21.3 (6.59) | 21.0 (5.92) | 0.518 | 0.468 | 0.610 | 0.925 |
| Females | 20.8 (10.2) | 17.7 (7.85) | | | | 0.401 |
| **Faecal output (g day$^{-1}$)** | | | | | | |
| Males | 5.20 (0.53) | 4.75 (0.81) | 0.653 | **0.0059** | 0.200 | 0.234 |
| Females | 4.11 (1.02) | 4.33 (0.58) | | | | 0.538 |

Note: Control group represents normobilirubinaemic heterozygote littermates. Gunn group represents hyperbilirubinaemic homozygote littermates. Adult rats were 14 weeks of age (females: control *n* = 9, Gunn *n* = 10; males: control *n* = 8, Gunn *n* = 9). Daily measures are an average of four separate days measured during the study. Values are represented as means (SD). Two-way ANOVA was performed with main effects: phenotype (Gunn or control) and sex (male or female). All *post hoc* analyses compared differences between phenotype within the same sex. Statistically significant (*P* < 0.05) *P* values are highlighted in bold.

were significantly greater in Gunn rats than controls, corroborating the idea that increased cholesterol synthesis is a counter-balancing mechanism to increased sterol excretion. Moreover, significant interaction was observed between sex and phenotype for faecal bile acid (day 14 interaction: *P* < 0.0001; day 18 interaction: *P* = 0.0005)

and total sterol (day 14 interaction: *P* < 0.0001; day 18 interaction: *P* = 0.0025) excretion at days 14 and 18, and at day 14 for faecal cholesterol excretion (interaction: *P* = 0.0248; Fig. 3). *Post hoc* analysis revealed that faecal bile acid (day 14: *P* < 0.0001; day 18: *P* < 0.0001), cholesterol (day 14: *P* = 0.0028; day 18: *P* = 0.0113)

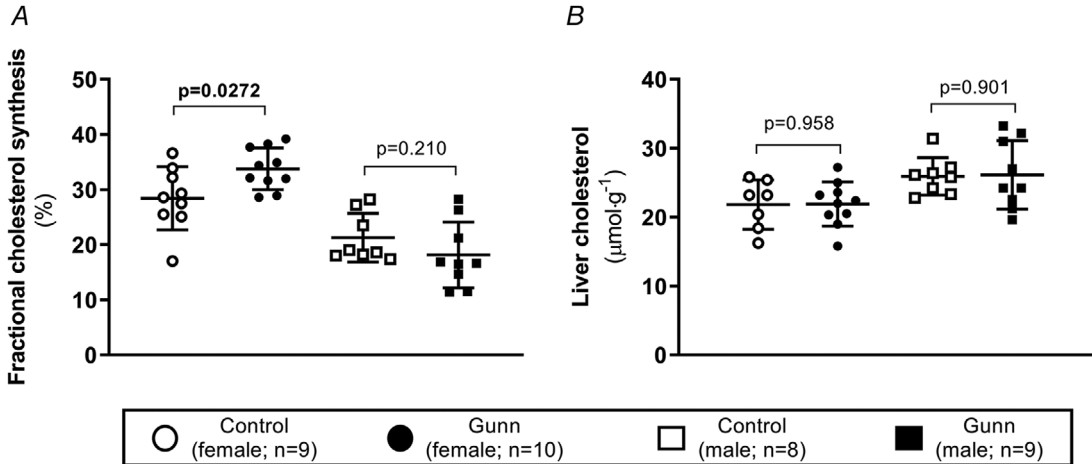

**Figure 1. *De novo* fractional cholesterol synthesis and hepatic cholesterol content of adult Gunn (hyperbilirubinaemic) and control (normobilirubinaemic) rats**
*A*, the rate of serum fractional cholesterol synthesis measured as a percentage (%) of newly formed $^{13}$C-cholesterol at steady state. *B*, total cholesterol content per gram of liver tissue. Data are presented as means (SD). Two-way ANOVA was performed with main effects: phenotype (Gunn or control) and sex (male or female). All *post hoc* analyses compared differences between phenotypes within the same sex. Statistically significant (*P* < 0.05) *P* values are highlighted in bold.

and total sterol excretion (day 14: $P < 0.0001$; day 18: $P < 0.0001$) were higher in female Gunn rats than controls, however, no difference existed between male groups (Fig. 3).

### Elevated net intestinal cholesterol flux in female Gunn rats

Faecal cholesterol excretion can be affected by four different mechanisms including: (1) dietary cholesterol intake; (2) biliary cholesterol secretion; (3) trans-intestinal cholesterol secretion (TICE); and (4) cholesterol reabsorption (Fig. 4). In this study, the diet did not contain

cholesterol, therefore, faecal cholesterol output was only determined by the latter three mechanisms. Since TICE increases while cholesterol reabsorption reduces the rate of faecal cholesterol excretion, their mutually opposing effects can be defined as a net (outward) intestinal cholesterol flux (TICE – cholesterol reabsorption; Fig. 4) (van de Peppel *et al.* 2019). Consequently, net intestinal cholesterol flux can be estimated by subtracting biliary cholesterol secretion from faecal cholesterol excretion. There was no effect of phenotype ($P = 0.185$) or inter-action ($P = 0.0931$) on the net (outward) intestinal cholesterol flux. However, *post hoc* analysis demonstrated that female Gunn rats had significantly elevated net

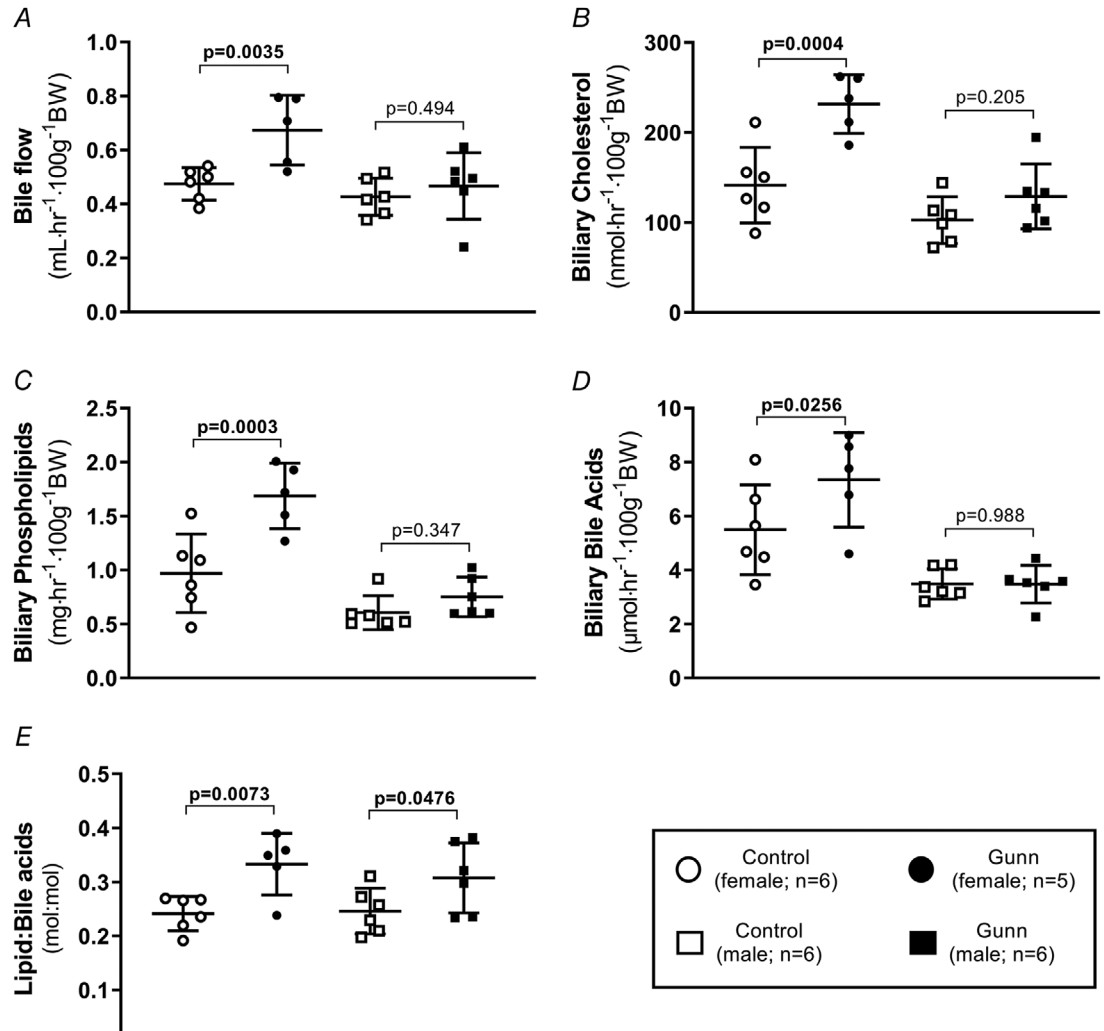

**Figure 2. Biliary lipid secretion of adult Gunn (hyperbilirubinaemic) and control (normobilirubinaemic) rats**
*A*, bile flow rate normalized for body weight. *B–D*, total cholesterol, phospholipids and bile acids secreted through bile normalized for body weight. *E*, biliary lipid (cholesterol + phospholipid) relative to bile acid secretion (mol:mol). Data are presented as means (SD). Two-way ANOVA was performed with main effects: phenotype (Gunn or control) and sex (male or female). All *post hoc* analyses compared differences between phenotypes within the same sex. Statistically significant ($P < 0.05$) *P* values are highlighted in bold.

intestinal cholesterol flux compared with controls ($P = 0.0427$; Fig. 4C).

### Altered bile acid composition in bile of adult female Gunn rats

Considering the enhanced biliary and faecal excretion of total bile acids in female Gunn rats, the biliary bile acid composition was assessed (proportion (%) of individual bile acid species related to total bile acid secretion). Relative secretion of TCA (phenotype: $P = 0.0087$) was significantly lower while TCDCA (phenotype: $P = 0.0226$), GCDCA (phenotype: $P = 0.0001$), TUDCA (phenotype: $P = 0.0007$) and GHDCA (phenotype: $P = 0.0396$) were significantly greater in Gunn rats (Fig. 5). There was also a significant interaction between sex and phenotype on relative secretion of TCA (interaction: $P = 0.0015$), GCDCA (interaction: $P = 0.0005$),

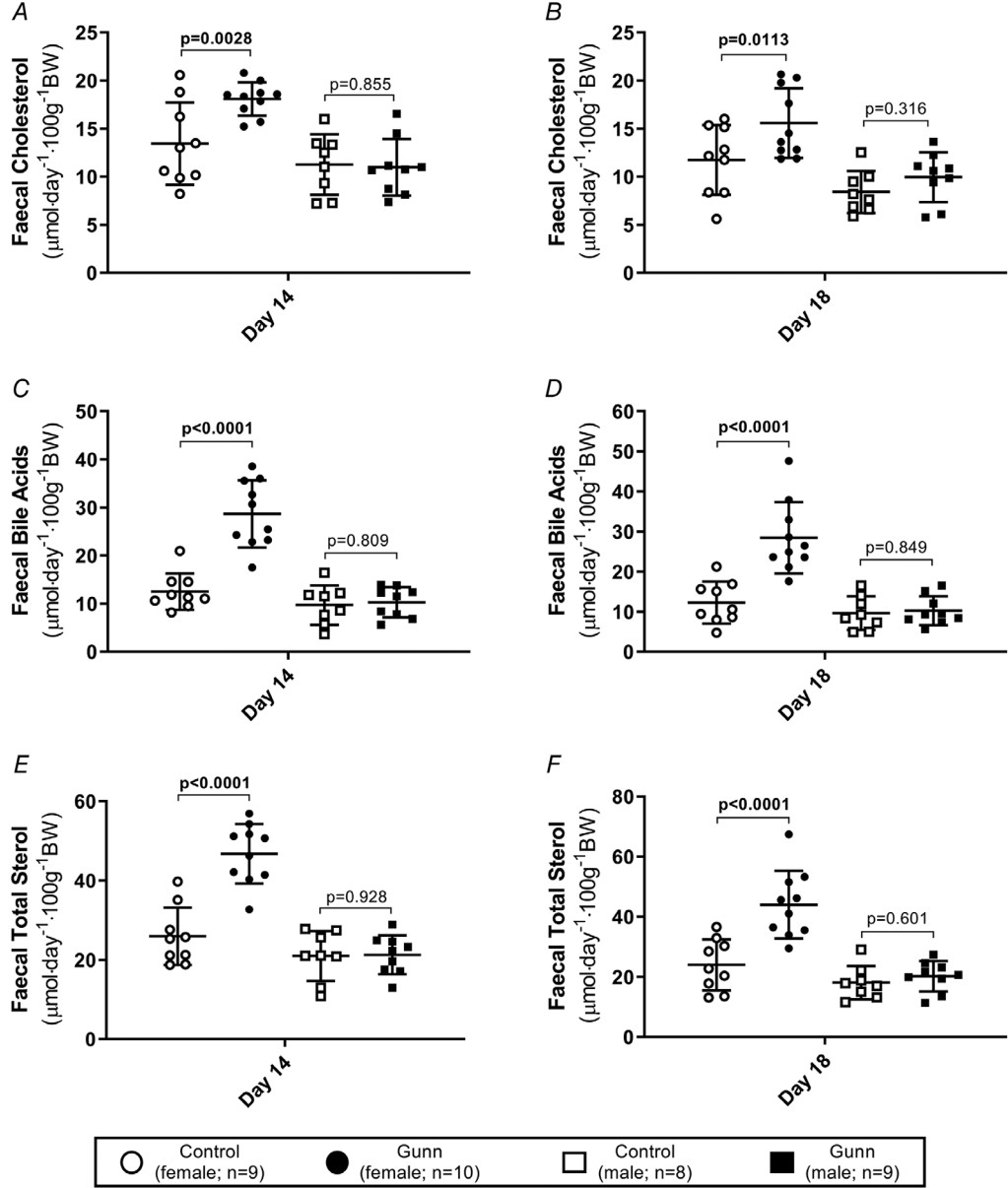

**Figure 3. Daily faecal sterol excreted, normalized for body weight of adult Gunn (hyperbilirubinaemic) and control (normobilirubinaemic) rats**
*A and B*, daily faecal excretion of cholesterol normalized for body weight. *C and D*, daily faecal excretion of bile acids normalized for body weight. *E and F*, daily faecal excretion of total sterols (cholesterol + bile acids) normalized for body weight. Data are presented as means (SD). Two-way ANOVA was performed with main effects: phenotype (Gunn or control) and sex (male or female). All *post hoc* analyses compared differences between phenotypes within the same sex. Statistically significant ($P < 0.05$) P values are highlighted in bold.

TUDCA (interaction: $P = 0.0002$) and GHDCA (interaction: $P = 0.0435$; Fig. 5). *Post hoc* analysis demonstrated that the proportion of TCA (*post hoc*: $P = 0.0002$) was significantly less while CA (*post hoc*: $P = 0.0468$), GCDCA (*post hoc*: $P < 0.0001$), TUDCA (*post hoc*: $P < 0.0001$) and GHDCA (*post hoc*: $P = 0.0070$) were significantly greater in female Gunn than control rats (Fig. 5). There was a significant interaction between sex and phenotype on the relative secretion of glycine (interaction: $P = 0.0327$) and taurine (interaction: $P = 0.0195$) conjugated bile acids (Fig. 6). *Post hoc* analysis showed that relative secretion

of glycine (*post hoc*: $P = 0.0179$) and taurine (*post hoc*: $P = 0.0047$) conjugated bile acids was greater and reduced, respectively, in female Gunn compared with control rats (Fig. 6).

## Perturbed expression of genes involved in cholesterol metabolism in female Gunn rats

To ascertain whether differences in circulating lipid concentrations and biliary/faecal lipid excretion were

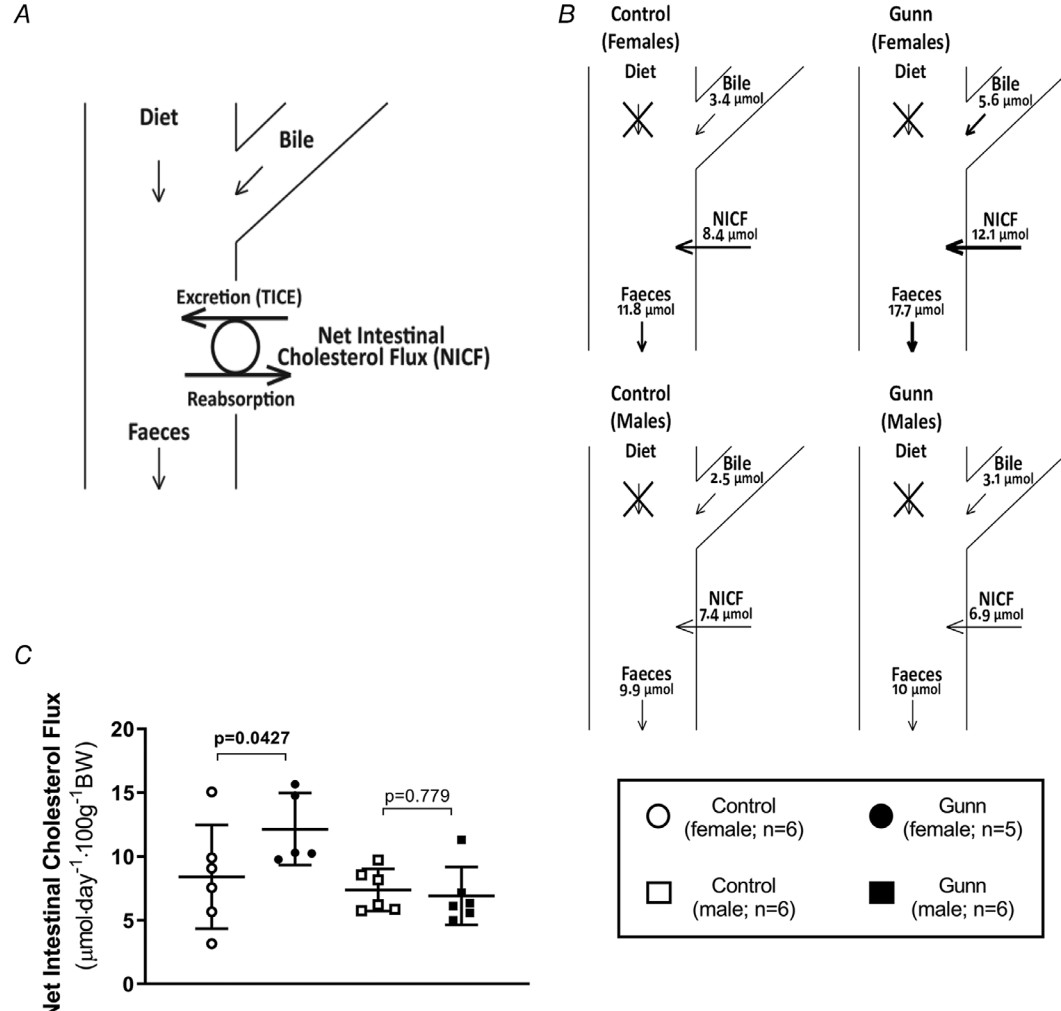

**Figure 4. Net intestinal cholesterol flux of adult Gunn (hyperbilirubinaemic) and control (normobilirubinaemic) rats**
*A*, model describing the four mechanisms (diet, biliary secretion, cholesterol reabsorption and TICE) that affect the rate of faecal cholesterol excretion. *B*, the daily rates of biliary cholesterol secretion, net intestinal cholesterol flux and faecal cholesterol excretion. Since the diet did not contain cholesterol, its contribution was disregarded. The net intestinal cholesterol flux was defined as the overall contribution of TICE and cholesterol reabsorption, and it was estimated by subtracting daily biliary cholesterol secretion from daily faecal cholesterol excretion. *C*, the daily net intestinal cholesterol flux in a graphical format. Data are presented as means (SD). Two-way ANOVA was performed with main effects: phenotype (Gunn or control) and sex (male or female). All *post hoc* analyses compared differences between phenotypes within the same sex. Statistically significant ($P < 0.05$) *P* values are highlighted in bold.

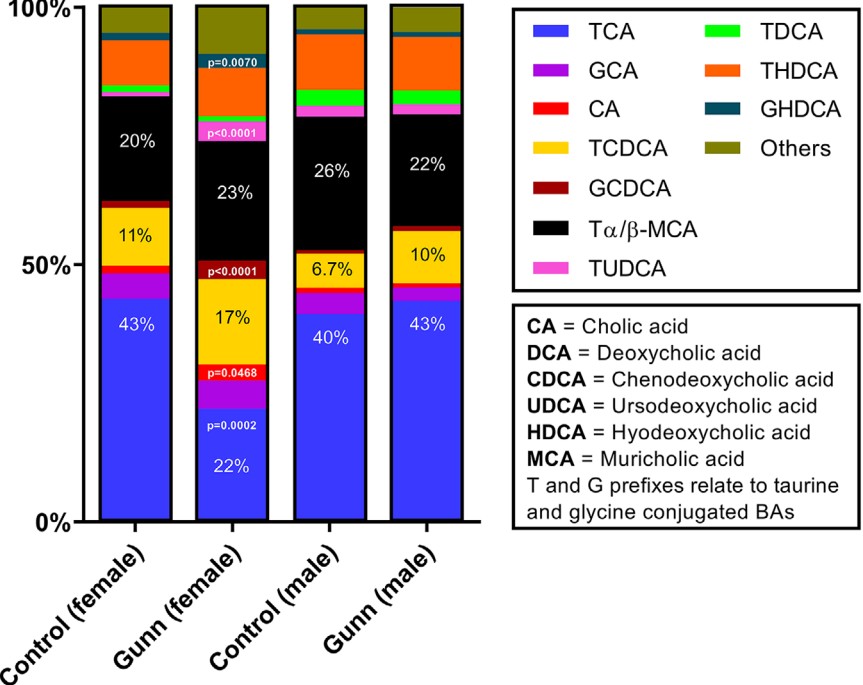

**Figure 5. Relative composition of biliary bile acid species of adult Gunn (hyperbilirubinaemic) and control (normobilirubinaemic) rats**

Bile acid species are presented as a mole percentage (%) of total bile acids excreted over an hour. 'Others' represents the sum contribution of A-MCA, GUDCA, CDCA, GDCA, TLCA, GLCA, $\beta$-MCA, O-MCA and HDCA. Two-way ANOVA was performed with main effects: phenotype (Gunn or control) and sex (male or female). All *post hoc* analyses compared differences between phenotypes within the same sex and only significant ($P < 0.05$) *P* values are reported on the figure. [Colour figure can be viewed at wileyonlinelibrary.com]

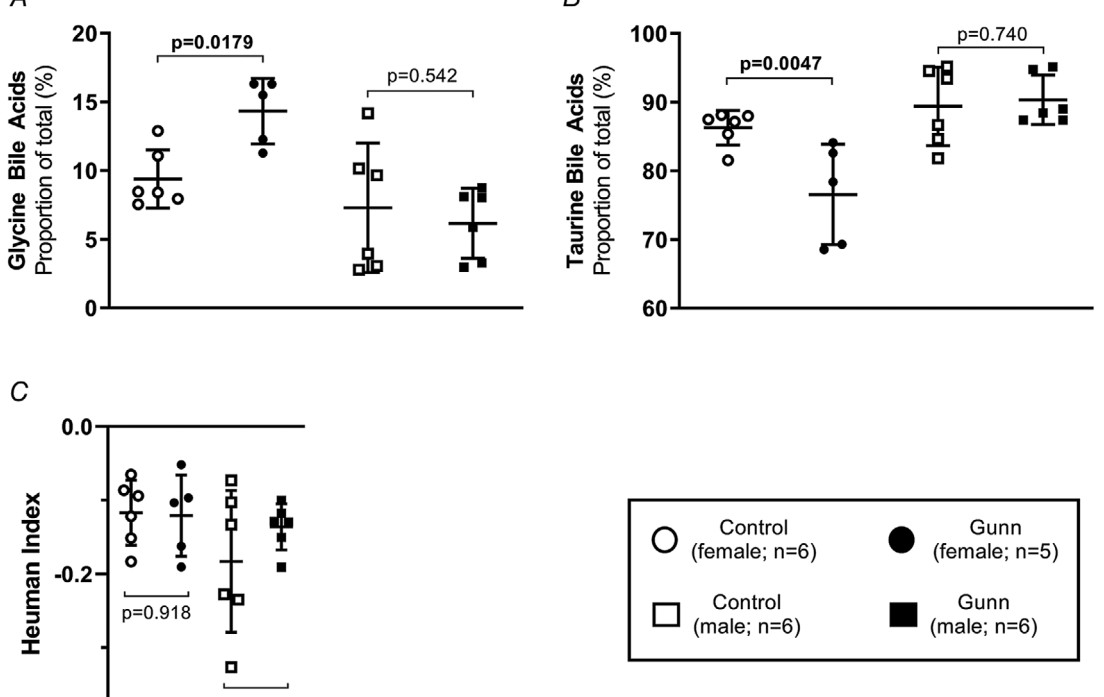

**Figure 6. Relative contribution of biliary glycine and taurine bile acid species and the hydrophobicity index of adult Gunn (hyperbilirubinaemic) and control (normobilirubinaemic) rats**

*A*, bile acid species are presented as a mole percentage (%) of total bile acids excreted over an hour. *B*, Heuman index of hydrophobicity of the biliary bile acid pool. Two-way ANOVA was performed with main effects: phenotype (Gunn or control) and sex (male or female). All *post hoc* analyses compared differences between phenotypes within the same sex. Statistically significant ($P < 0.05$) *P* values are highlighted in bold.

associated with the expression of genes regulating lipid metabolism and transport, RT-qPCR analysis was performed on liver tissue of female adult rats (Fig. 7 and Table 4). The expression of genes regulating several enzymes within the cholesterol synthesis pathway, including *Pmvk* ($P = 0.0411$) and *Cyb5r3* ($P = 0.0288$), was significantly decreased (Fig. 7*A*) while expression of *Hmgcr*, the rate-limiting enzyme of cholesterol synthesis,

was not different between groups (1.04-fold increase in female Gunn rats *vs.* controls; $P = 0.554$; Table 4). Furthermore, several genes involved in lipoprotein transport were differentially expressed between groups. *Lrp6* ($P = 0.0482$) and lipoprotein-associated proteins, *Colec12* ($P = 0.0409$) and *Apof* ($P = 0.0274$), were significantly reduced, while *Ldlr* ($P = 0.0105$) was significantly upregulated in female Gunn rats *versus* controls (Fig. 7*A*). Additionally, the gene responsible for cholesterol esterification (*Soat2*) was significantly increased in female Gunn rats compared with controls ($P = 0.0378$; Fig. 7*A*). Finally, genes that tended to differ between groups ($P < 0.15$) are shown in Fig. 7*B*, while the remaining genes that were not significantly affected are reported in Table 4.

## Altered expression of proteins involved in cholesterol metabolism in female Gunn rats

To determine whether differences in gene expression were related to changes in protein expression, western blot analysis was undertaken on key proteins involved in cholesterol metabolism and lipoprotein transport. Expression of CYB5R3 ($P = 0.0276$) was reduced while HMGCR expression was non-significantly elevated ($P = 0.0794$) in female Gunn rats *versus* controls. No difference in the expression of these proteins was observed in male animals (Fig. 8*A-B*). CYP7A1 expression, the rate-limiting enzyme of the classical pathway of bile acid synthesis, was significantly greater in male Gunn rats *versus* controls ($P = 0.0170$; Fig. 8*C*). In agreement with gene expression analysis, expression of LDLr was significantly upregulated in female Gunn rats ($P = 0.0018$) with no difference in male rats ($P = 0.582$; Fig. 8*D*). ABCA1 expression was not affected across groups (Fig. 8*E*). The nuclear form of SREBP2, a master regulator of genes that regulate cholesterol metabolism, was significantly greater in female Gunn rats than in controls ($P = 0.0037$; Fig. 9). SREBP2 was not measured in male rats because there was no difference in down-stream targets, including the fractional rate of cholesterol synthesis and LDLr expression, between groups.

## Discussion

Hyperbilirubinaemia in humans and animal models is associated with reduced circulating cholesterol concentrations. However, an explanation for this relationship remains unknown (Bulmer *et al.* 2013). Determining the underlying mechanisms causing the hypolipidaemic state is critical to revealing the physiological importance and therapeutic potential of bilirubin/UGT1A1 dysfunction. The main findings of this study included an increased faecal sterol excretion

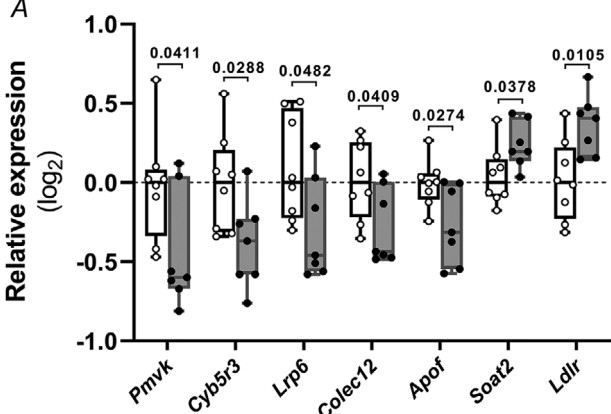

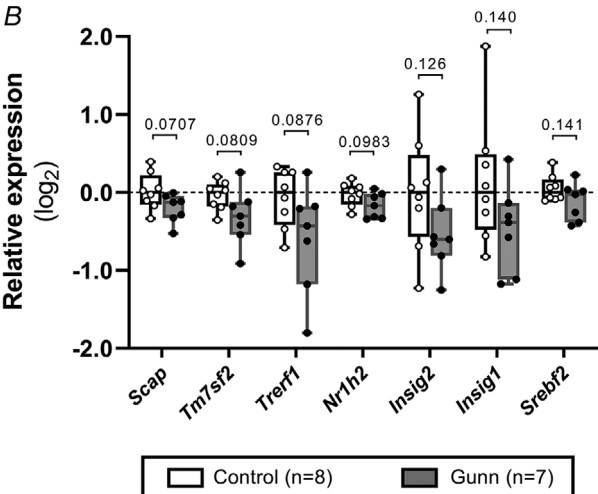

**Figure 7. Hepatic gene expression of female adult Gunn (hyperbilirubinaemic) and control (normobilirubinaemic) rats**
*A*, genes that were significantly different between groups ($P < 0.05$). *B*, genes that tended ($P < 0.10$) to demonstrate differences. Abbreviations: *Pmvk*, phosphomevalonate kinase; *Cyb5r3*, NADH-cytochrome b5 reductase 3; *Lrp6*, low-density lipoprotein receptor-related protein 6; *Colec12*, collectin12; *Apof*, apolipoprotein F; *Soat2*, sterol O-acyltransferase 2; *Ldlr*, low-density lipoprotein receptor; *Scap*, Srebf cleavage activating protein; *Tm7sf2*, transmembrane 7 subfamily member 2; *Trerf1*, transcriptional regulating factor 1; *Nr1h2*, nuclear receptor subfamily 1, group H, member 2; *Insig1-2*, insulin-induced gene 1–2; *Srebf2*, sterol regulatory element-binding factor 2. A two-tailed unpaired *t* test was used for statistical analysis. Fold change data from Table 4 were transformed to log$_2$ for graphical purposes. Data are presented as medians ± range.

**Table 4. Hepatic gene expression of female adult Gunn (hyperbilirubinaemic; *n* = 7) and control (normobilirubinaemic; *n* = 8) rats.
Fold change expressed relative to controls**

| Gene name | Gene abbreviation | Fold change [IQR] | *P* value |
|---|---|---|---|
| *ATP-binding cassette, subfamily A (ABC1), member 1* | *Abca1* | 1.02 [0.43] | 0.455 |
| *ATP-binding cassette, subfamily A (ABC1), member 2* | *Abca2* | 1.05 [0.15] | 0.310 |
| *ATP-binding cassette, subfamily G (WHITE), member 1* | *Abcg1* | 0.88 [0.40] | 0.548 |
| *Aldo-keto reductase family 1, member D1* | *Akr1d1* | 1.23 [0.36] | 0.363 |
| *Angiopoietin-like 3* | *Angptl3* | 1.11 [0.26] | 0.994 |
| *Ankyrin repeat, family A (RFXANK-like), 2* | *Ankra2* | 0.91 [0.46] | 0.625 |
| *Apolipoprotein A-I* | *Apoa1* | 0.95 [0.25] | 0.231 |
| *Apolipoprotein A-II* | *Apoa2* | 0.97 [0.06] | 0.732 |
| *Apolipoprotein A-IV* | *Apoa4* | 0.69 [0.17] | 0.618 |
| *Apolipoprotein B* | *Apob* | 0.98 [0.28] | 0.294 |
| *Apolipoprotein C-I* | *Apoc1* | 0.91 [0.13] | 0.333 |
| *Apolipoprotein C-III* | *Apoc3* | 0.85 [0.36] | 0.509 |
| *Apolipoprotein E* | *Apoe* | 0.99 [0.38] | 0.548 |
| *Apolipoprotein F* | *Apof* | 0.80 [0.25] | **0.0274** |
| *Apolipoprotein L, 2* | *Apol2* | 0.57 [0.31] | 0.486 |
| *Cadherin 13* | *Cdh13* | 1.04 [0.51] | 0.948 |
| *Carboxyl ester lipase* | *Cel* | 0.80 [0.63] | 0.812 |
| *Chymotrypsin-like elastase family, member 3B* | *Cela3b* | 0.84 [0.71] | 0.850 |
| *CCHC-type zinc finger, nucleic acid binding protein* | *Cnbp* | 0.82 [0.37] | 0.355 |
| *Collectin subfamily member 12* | *Colec12* | 0.74 [0.23] | **0.0409** |
| *Chemokine (C-X-C motif) ligand 16* | *Cxcl16* | 0.84 [0.19] | 0.437 |
| *Cytochrome b5 reductase 3* | *Cyb5r3* | 0.77 [0.17] | **0.0288** |
| *Cytochrome P450, family 11, subfamily a, polypeptide 1* | *Cyp11a1* | 0.68 [0.83] | 0.274 |
| *Cytochrome P450, family 39, subfamily a, polypeptide 1* | *Cyp39a1* | 1.01 [0.48] | 0.884 |
| *Cytochrome P450, family 46, subfamily a, polypeptide 1* | *Cyp46a1* | 0.90 [0.77] | 0.545 |
| *Cytochrome P450, family 51* | *Cyp51* | 0.79 [0.37] | 0.173 |
| *Cytochrome P450, family 7, subfamily a, polypeptide 1* | *Cyp7a1* | 1.23 [1.69] | 0.536 |
| *Cytochrome P450, family 7, subfamily b, polypeptide 1* | *Cyp7b1* | 0.74 [0.24] | 0.216 |
| *24-dehydrocholesterol reductase* | *Dhcr24* | 1.19 [0.36] | 0.382 |
| *7-dehydrocholesterol reductase* | *Dhcr7* | 1.07 [0.29] | 0.655 |
| *Emopamil binding protein (sterol isomerase)* | *Ebp* | 0.81 [0.31] | 0.243 |
| *Farnesyl diphosphate farnesyl transferase 1* | *Fdft1* | 0.84 [0.44] | 0.280 |
| *Farnesyl diphosphate synthase* | *Fdps* | 0.89 [0.32] | 0.164 |
| *High-density lipoprotein binding protein* | *Hdlbp* | 0.97 [0.17] | 0.967 |
| *3-hydroxy-3-methylglutaryl-Coenzyme A reductase* | *Hmgcr* | 1.04 [1.34] | 0.554 |
| *3-hydroxy-3-methylglutaryl-Coenzyme A synthase 1* | *Hmgcs1* | 0.86 [0.17] | 0.649 |
| *Isopentenyl-diphosphate delta isomerase 1* | *Idi1* | 0.80 [0.27] | 0.912 |
| *Interleukin 4* | *Il4* | 0.78 [0.23] | 0.613 |
| *Insulin-induced gene 1* | *Insig1* | 0.66 [0.17] | 0.140 |
| *Insulin-induced gene 2* | *Insig2* | 0.77 [0.29] | 0.126 |
| *Lecithin cholesterol acyltransferase* | *Lcat* | 0.89 [0.30] | 0.315 |
| *Low-density lipoprotein receptor* | *Ldlr* | 1.32 [0.21] | **0.0105** |
| *Low-density lipoprotein receptor adaptor protein 1* | *Ldlrap1* | 0.95 [0.14] | 0.737 |
| *Leptin* | *Lep* | 0.64 [0.76] | 0.335 |
| *Lipase, hormone sensitive* | *Lipe* | 1.14 [0.13] | 0.453 |
| *Low-density lipoprotein receptor-related protein 10* | *Lrp10* | 1.02 [0.04] | 0.937 |
| *Low-density lipoprotein-related protein 12* | *Lrp12* | 1.14 [0.34] | 0.739 |
| *Low-density lipoprotein receptor-related protein 6* | *Lrp6* | 0.73 [0.27] | **0.0482** |
| *Low-density lipoprotein receptor-related protein associated protein 1* | *Lrpap1* | 0.91 [0.32] | 0.343 |
| *Membrane-bound transcription factor peptidase, site 1* | *Mbtps1* | 1.02 [0.25] | 0.690 |
| *Mevalonate (diphospho) decarboxylase* | *Mvd* | 1.13 [0.41] | 0.844 |
| *Mevalonate kinase* | *Mvk* | 1.05 [0.31] | 0.880 |

*(Continued)*

**Table 4. (Continued)**

| Gene name | Gene abbreviation | Fold change [IQR] | *P* value |
|---|---|---|---|
| *Nuclear receptor subfamily 0, group B, member 2* | Nr0b2 | 1.56 [1.00] | 0.894 |
| *Nuclear receptor subfamily 1, group H, member 2* | Nr1h2 | 0.89 [0.17] | 0.0983 |
| *Nuclear receptor subfamily 1, group H, member 3* | Nr1h3 | 0.98 [0.17] | 0.853 |
| *Nuclear receptor subfamily 1, group H, member 4* | Nr1h4 | 1.00 [0.52] | 0.663 |
| *NAD(P) dependent steroid dehydrogenase-like* | Nsdhl | 0.77 [0.42] | 0.226 |
| *Oxidized low-density lipoprotein (lectin-like) receptor 1* | Olr1 | 0.93 [0.27] | 0.689 |
| *Oxysterol binding protein-like 1A* | Osbpl1a | 1.00 [0.13] | 0.537 |
| *Oxysterol binding protein-like 5* | Osbpl5 | 0.88 [0.19] | 0.622 |
| *Proprotein convertase subtilisin/kexin type 9* | Pcsk9 | 1.10 [0.38] | 0.628 |
| *Phosphomevalonate kinase* | Pmvk | 0.66 [0.21] | **0.0411** |
| *Protein kinase, AMP-activated, alpha 1 catalytic subunit* | Prkaa1 | 0.77 [0.45] | 0.361 |
| *Protein kinase, AMP-activated, alpha 2 catalytic subunit* | Prkaa2 | 1.07 [0.12] | 0.678 |
| *Protein kinase, AMP-activated, gamma 2 non-catalytic subunit* | Prkag2 | 0.99 [0.12] | 0.756 |
| *SREBF chaperone* | Scap | 0.92 [0.14] | 0.0707 |
| *Scavenger receptor class F, member 1* | Scarf1 | 0.95 [0.26] | 0.696 |
| *Sorting nexin 17* | Snx17 | 0.99 [0.19] | 0.456 |
| *Sterol O-acyltransferase 1* | Soat1 | 0.71 [0.29] | 0.163 |
| *Sterol O-acyltransferase 2* | Soat2 | 1.14 [0.14] | **0.0378** |
| *Sortilin-related receptor, LDLR class A repeats-containing* | Sorl1 | 1.06 [0.40] | 0.575 |
| *Sterol regulatory element-binding transcription factor 1* | Srebf1 | 0.84 [0.66] | 0.651 |
| *Sterol regulatory element-binding transcription factor 2* | Srebf2 | 0.98 [0.22] | 0.141 |
| *Stabilin 2* | Stab2 | 0.60 [0.54] | 0.372 |
| *StAR-related lipid transfer (START) domain containing 3* | Stard3 | 0.92 [0.16] | 0.874 |
| *Transmembrane 7 superfamily member 2* | Tm7sf2 | 0.81 [0.18] | 0.0809 |
| *Transcriptional regulating factor 1* | Trerf1 | 0.74 [0.33] | 0.0876 |
| *Very low-density lipoprotein receptor* | Vldlr | 1.05 [0.41] | 0.973 |

Note: Control group represents normobilirubinaemic heterozygote littermates. Gunn group represents hyperbilirubinaemic homozygote littermates. Adult rats were 14 weeks of age (females: control *n* = 8, Gunn *n* = 7). A two-tailed unpaired *t* test was used for statistical analysis. Data are presented as median fold changes and interquartile ranges (IQR) compared with controls. Statistically significant (*P* < 0.05) *P* values are highlighted in bold.

in female hyperbilirubinaemic Gunn rats which was associated with elevated fractional cholesterol synthesis, greater hepatic LDLr expression and profound hypocholesterolaemia. Surprisingly, the effects of UGT1A1 impairment/hyperbilirubinaemia on sterol metabolism were mostly specific to female animals with only minor effects in male Gunn rats. Increased faecal sterol excretion may explain the negative relationship between bilirubin and circulating cholesterol in hyperbilirubinaemic animals and justifies further investigation in humans with GS.

Adult female Gunn rats demonstrated a >50% reduction in serum cholesterol concentrations compared with controls, largely due to a reduction in HDL-C which is the dominant circulating lipoprotein in rodents (Camus *et al.* 1983). These findings are congruent with previous reports of lower circulating cholesterol in other colonies of female Gunn rats (Boon *et al.* 2012; Wallner *et al.* 2013). In contrast, cholesterol concentrations were not different in adult male rats, which is in opposition to the

results of Wallner *et al.* (2013), who reported a ∼20% reduction in serum cholesterol in male Gunn rats. This difference in results may be explained by the younger age of the animals studied by Wallner *et al.* (2013) (∼8 *vs.* 16 weeks of age). Indeed, we found a significant interaction between sex and age in Gunn rats where we observed ∼30% lower serum cholesterol concentrations in male juvenile rats (3–4 weeks of age) but not in females. A novel finding of this study was that adult female Gunn rats had lower circulating phospholipids. Therefore, changes to circulating lipid concentrations suggest involvement of the liver since it tightly regulates the total body sterol pool (Dietschy & Turley, 2002; Van Der Velde *et al.* 2010). Moreover, these data broadly suggest that age and sex may play an important role in various aspects of lipid metabolism in hyperbilirubinaemic rats.

Previous studies show that lipid-lowering therapies, including ezetimibe, do not affect the total sterol pool despite reducing circulating cholesterol levels by increasing faecal sterol excretion (Dietschy & Turley, 2002;

Jakulj *et al.* 2016; van de Peppel *et al.* 2019). Ezetimibe reduces circulating cholesterol by increasing faecal cholesterol excretion, however, this does not change the sterol pool because of a compensatory increase in cholesterol synthesis (Jakulj *et al.* 2016; van de Peppel *et al.* 2019). Therefore, reduced serum cholesterol concentrations in adult female Gunn rats may be caused indirectly, via a change in sterol excretion. Cholesterol is primarily removed intact or after conversion to bile acids through the gastrointestinal tract as faeces (Van Der Velde *et al.* 2010). Female Gunn rats demonstrated a 33–35% increase in faecal cholesterol output and a more than twofold increase in faecal bile acid excretion compared with controls. Therefore, changes to faecal sterol output must be counterbalanced by dietary intake and/or endogenous synthesis to maintain a constant cholesterol

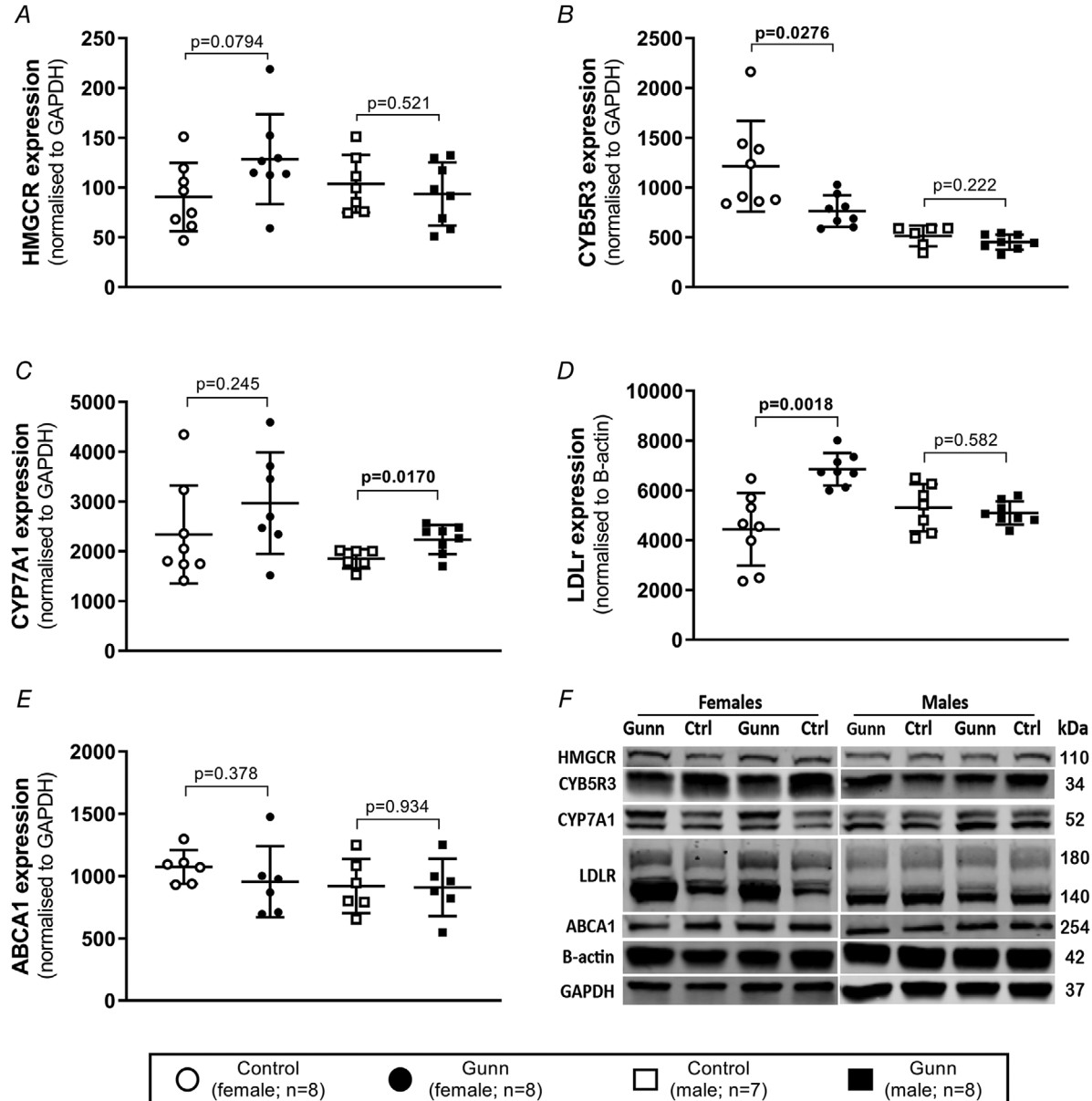

**Figure 8. Protein expression analysis using western blot for targets of cholesterol synthesis, transport and breakdown in livers from adult Gunn (hyperbilirubinaemic) and control (normobilirubinaemic) rats**
*A–E*, HMGCR, CYB5R3, CYP7A1 and ABCA1 expression normalized to GAPDH expression and LDLr expression normalized to B-actin expression. *F*, cropped image that represents a single western blot run. Abbreviations: HMGCR, 3-hydroxy-3-methyl-glutaryl-coenzyme A reductase; CYB5R3, NADH-cytochrome b5 reductase 3; CYP7A1, cholesterol 7 alpha-hydroxylase; LDLr, low-density lipoprotein receptor; ABCA1, ATP-binding cassette transporter; Data are presented as means (SD). A two-tailed unpaired *t* test was used for statistical analysis. Statistically significant ($P < 0.05$) *P* values are highlighted in bold.

pool. Considering that the diet consumed by the rats in this study did not contain cholesterol, the sterols lost through faeces must be solely compensated for by endogenous synthesis (Schonewille *et al.* 2016; van de Peppel *et al.* 2019). Consistent with this, fractional cholesterol synthesis was elevated in female Gunn rats counteracting increased sterol excretion to maintain cholesterol balance.

Although this study clearly describes the functional impact of hyperbilirubinaemia/UGT1A1 dysfunction on sterol excretion, the mechanism explaining this effect remains to be elucidated. Nevertheless, increased bile acid excretion can only be attributed to one of three mechanisms including: (1) increased biliary bile acid secretion; (2) decreased intestinal bile acid reabsorption; or (3) a combination of 1 and 2 (de Boer *et al.* 2018). Even though biliary bile acid secretion was elevated by 33% in adult female Gunn rats this cannot account for the more than twofold higher faecal bile acid excretion compared with female control rats. Therefore, it can be assumed that the greater faecal bile acid excretion in Gunn rats is primarily caused by reduced intestinal reabsorption. Studies investigating the impact of disrupted intestinal bile acid reabsorption on sterol metabolism report similar findings observed in female Gunn rats. For instance, ABST$^{-/-}$ mice with impaired intestinal bile acid reabsorption demonstrate a more than twofold increase in faecal sterol excretion with an increase in *in vivo* cholesterol synthesis and greater gene expression of *Cyp7a1*, the rate-limiting enzyme of bile acid synthesis (van de Peppel *et al.* 2019). Increased bile acid synthesis is a physiological response to offset the depleted bile acid pool in conditions of elevated bile acid excretion (van de Peppel *et al.* 2019). Similar results are found in animal studies of complete bile diversion (Smit *et al.* 1990) and in animals/humans treated with bile

acid-sequestering medication that enhance faecal bile acid loss (Einarsson *et al.* 1991). Given that bile acid excretion equals bile acid synthesis at steady state (Chiang, 2013), adult female Gunn rats must experience increased bile acid synthesis. This conclusion is partly supported by a non-significant ($P = 0.245$) 27% greater hepatic CYP7A1 protein expression in adult female Gunn rats compared with controls. Nevertheless, future investigations are required to confirm that bile acid reabsorption is reduced in female Gunn rats and that this is the primary cause of increased faecal bile acid excretion.

Intriguingly, adult female Gunn rats also show greater biliary cholesterol secretion. It is tempting to speculate that increased cholesterol synthesis and/or increased HDL-C uptake are involved in this observation (Dikkers & Tietge, 2010). Biliary cholesterol can predominantly originate from HDL-C and the rate of hepatic HDL-C uptake could impact the rate of biliary cholesterol secretion (Dikkers & Tietge, 2010). The profoundly reduced serum HDL-C concentrations in female Gunn rats may be related to reduced HDL production, increased clearance, or both. It seems less likely that female Gunn rats had reduced HDL-C production because hepatic ABCA1 hepatic protein and *Apoa1* gene (Table 4) expression were unchanged. Rather, increased hepatic HDL-C uptake could be related to the increased biliary cholesterol secretion in female Gunn rats (Dikkers & Tietge, 2010). Alternatively, biliary cholesterol secretion may have been increased in female Gunn rats due to increased coupling of lipids to bile acid secretion due to reduced biliary concentration of bilirubin conjugates (Bulmer *et al.* 2013). Conjugated bilirubin and other organic anions dissociate biliary lipid secretion from bile acids and cause a reduction in overall biliary output of cholesterol and phospholipids (Verkade, 2000). However,

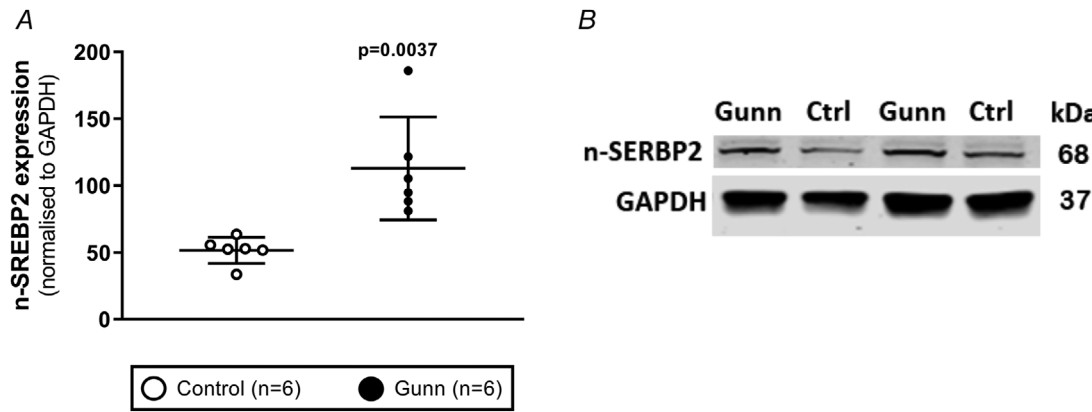

**Figure 9. Protein expression analysis using western blot for nuclear form of SREBP2 (n-SREBP2) in livers from adult female Gunn (hyperbilirubinaemic) and control (normobilirubinaemic) rats**
*A*, SREBP2 expression normalized to GAPDH expression. *B*, an example cropped image that represents a single western blot run. Abbreviations: SREBP2, sterol regulatory element-binding protein 2. Data are presented as means (SD). Statistical analyses were conducted using unpaired *t* tests. Statistically significant ($P < 0.05$) *P* values are highlighted in bold.

these studies commonly infused supra-physiological concentrations of organic anions. Thus, it is unknown whether increased coupling occurs at physiological or sub-physiological levels of organic anions in bile. The current study presents the first evidence of increased coupling of biliary lipids to bile acid secretion in Gunn rats that secrete substantially lower amounts of bilirubin conjugates in bile. Adult Gunn rats demonstrated greater relative secretion of biliary lipids to bile acids in part explaining the higher biliary cholesterol and phospholipid secretion in females.

Hepatic gene and protein expression was assessed to explore the impact of increased faecal sterol excretion on cholesterol metabolism and hypocholesterolaemia in female Gunn rats. Hepatic LDLr expression was elevated in female Gunn rats; a finding which is compatible with hypocholesterolaemia and increased cholesterol uptake by the liver (Daniels *et al.* 2009). Furthermore, female Gunn rats demonstrated elevated protein expression of the nuclear form of SREBP2, which is a master regulator of cholesterol metabolism that acts as a sterol-sensing transcription factor (Eberlé *et al.* 2004). During decreased intracellular cholesterol levels, endoplasmic reticulum-bound SREBP2 is cleaved from the SREBP2-SCAP-INSIG complex and translocates to the nucleus where it transactivates genes involved in cholesterol synthesis and transport (Eberlé *et al.* 2004). SREBP2 positively regulates expression of *Hmgcr*, *Ldlr*, *Pmvk*, *Hmgcs*, *Abcg5* and *Abcg8* (Horton *et al.* 2003). Therefore, increased *Ldlr* expression in female Gunn rats is consistent with greater expression of the nuclear form of SREBP2. However, no significant change in *Hmgcr* or *Hmgcs* gene expression occurred, while *Pmvk* gene expression was significantly decreased (Fig. 7). Although expression of *Abcg5* and *Abcg8* was not measured, female Gunn rats demonstrated greater biliary cholesterol secretion, indicating that the expression of these genes may be increased because they mediate the export of hepatic sterols into the bile canaliculus (Boyer, 2013).

Gene expression data of selected targets was validated using protein expression. Hepatic HMGCR protein expression tended to increase by ∼40% ($P = 0.0794$) in female Gunn rats, consistent with increased SREBP2 protein expression and cholesterol synthesis in the same animals (Van Der Wulp *et al.* 2013). Therefore, greater (rate-limiting) HMGCR expression potentially explains the elevated rate of cholesterol synthesis in female Gunn rats (DeBose-Boyd, 2008). Surprisingly, gene expression of *Hmgcr* was not elevated, suggesting that increased protein expression may occur due to reduced post-translational degradation of HMGCR in female Gunn rats mediated by reduced gene expression of *Insig1* and *Insig2* ($P < 0.15$; Fig. 7B) (DeBose-Boyd, 2008; Van Der Wulp *et al.* 2013). Taken together, these results suggest that female Gunn rats have increased hepatic

uptake of circulating cholesterol and elevated cholesterol synthesis, and this could be mediated through enhanced nuclear expression of SREBP2. However, to confirm this, future studies need to investigate the expression of other important cholesterol uptake receptors such as scavenger receptor class B type 1 and transcription factors such as SREBP1a. Nevertheless, this pattern of gene/protein regulation is a typical physiological response to reduced hepatic cholesterol content that aims to restore intracellular cholesterol as reported with lipid-lowering therapies (Schonewille *et al.* 2016).

Hepatic cholesterol concentrations are also affected by the rate of intestinal cholesterol absorption. Increased faecal cholesterol excretion in adult female Gunn rats suggests decreased intestinal reabsorption of cholesterol (Jakulj *et al.* 2010). Interestingly, bile acids are vital for efficient cholesterol absorption and their effectiveness varies depending on their physiochemical properties (de Boer *et al.* 2018). Wang *et al.* (2003) demonstrated that the hydrophobicity index of the biliary bile acid composition (Heuman index) was strongly and positively related to intestinal cholesterol absorption (Heuman *et al.* 1989; Wang *et al.* 2003). In comparison to hydrophilic bile acids (e.g. $\alpha/\beta$-MCA species), hydrophobic species such as CA and TCA are potent inducers of cholesterol absorption (Wang *et al.* 2003). A profoundly lower proportion of TCA was measured in the bile of female Gunn rats (minor elevation in MCA species). The reason for the difference in the proportions of bile acid species in female Gunn rats remains unknown; however, the over-all hydrophobicity index did not change. These data suggest that the difference in biliary bile acid composition in female Gunn rats was unlikely to affect cholesterol reabsorption.

Intestinal cholesterol reabsorption is also affected by the rate of bile acid reabsorption. In adult female Gunn rats, a profound increase in faecal bile acid excretion occurred, strongly suggesting reduced intestinal reabsorption of bile acids. ASBT$^{-/-}$ mice with reduced intestinal bile acid reabsorption also demonstrate a substantial decrease in cholesterol reabsorption underpinning the co-dependence of these two pathways (van de Peppel *et al.* 2019). Indeed, female Gunn rats had elevated net intestinal cholesterol flux which can only be a result of reduced intestinal cholesterol reabsorption and/or by increased TICE (van de Peppel *et al.* 2019). Consequently, it is tempting to speculate that cholesterol reabsorption is reduced in female Gunn rats. However, it is not possible to separate the individual contributions of the two mechanisms. Collectively, the findings of this study support a conclusion that perturbation in cholesterol balance in female Gunn rats is caused by increased excretion of bile acids and to a lesser extent of cholesterol. The resulting negative intestinal sterol balance is compensated for by elevated cholesterol synthesis.

Hepatic UGT1A1 is an important enzyme for $17\beta$-oestradiol excretion. UCB competes with $17\beta$-oestradiol for conjugation by UGT1A1 (Zhou *et al.* 2011; Sambasivarao, 2013); therefore, UGT1A1 dysfunction in Gunn rats possibly reduces $17\beta$-oestradiol excretion. Although circulating hormone concentrations were not measured in this study, it is plausible that female Gunn rats contain elevated levels of $17\beta$-oestradiol. The impact of oestrogen on lipid metabolism is well established in both humans and animals (Palmisano *et al.* 2017). Ovariectomized animals with very low oestrogen levels are heavier and have substantially higher serum cholesterol concentrations which are reversed by treatment with oestrogen, establishing a causal relationship of oestrogen with respect to lipid metabolism (Zhu *et al.* 2013). Interestingly, reduced circulating cholesterol concentrations following oestrogen treatment in rats are associated with increased LDLr expression suggesting that oestrogen induces hepatic cholesterol uptake (Koopen *et al.* 1999). Collectively, elevated oestrogen concentrations may potentially explain the hypocholesterolaemia and enhanced LDLr expression in female Gunn rats and this requires further investigation.

Considering that male animals are unlikely to experience the confounding effects of oestrogen metabolism, the effect of UGT1A1 inhibition and hyper-bilirubinaemia, should be clearer in these animals. Under these conditions, effects on lipid metabolism appeared to be milder in male Gunn rats. Although adult male Gunn rats demonstrated significantly lower body mass there was no change in serum cholesterol or any measures of sterol metabolism. Conversely, lower serum cholesterol was found in juvenile male Gunn rats, indicating that age plays an important factor in sterol metabolism in Gunn rats. In juvenile male Gunn rats, different mechanisms from those in females must be responsible for reduced cholesterol concentrations. These potentially relate to bilirubin more specifically. Bilirubin is an endogenous ligand for the AhR and PPAR$\alpha$, which are ligand-activated transcription factors that regulate lipid metabolism (Phelan *et al.* 1998; Stec *et al.* 2016). In our previous study we found that downstream targets of PPAR$\alpha$ were specifically upregulated in adult male, but not in female, Gunn rats (Vidimce *et al.* 2021). This suggests that bilirubin regulates lipid metabolism through PPAR$\alpha$ in male Gunn rats, but an alternative mechanism must be responsible for the changes observed in females. However, unlike studies with hyperbilirubinaemic mice that show PPAR$\alpha$-mediated decreases in adiposity (Hinds *et al.* 2017), minimal changes to fat metabolism occur in male Gunn rats (Vidimce *et al.* 2021).

Alternatively, bilirubin could modulate lipid metabolism by binding to AhR. Ligand activation of AhR causes repression of several genes involved in cholesterol metabolism such as *Hmgcr*, *Hmgcs* and *Pmvk* (Tanos *et al.* 2012). It is yet to be demonstrated whether AhR is activated *in vivo* in hyperbilirubinaemic animal models, although preliminary evidence shows that down-stream genes of AhR are upregulated in juvenile but not in adult Gunn rats (Kapitulnik & Gonzalez, 1993). The mechanisms responsible for the changes to lipid metabolism in Gunn rats remain unknown. However, they could include bilirubin agonism of PPAR$\alpha$ and AhR, and/or elevated oestrogen concentrations due to UGT1A1 impairment. Future studies are required to explore the importance of these receptors and of oestrogen in the regulation of lipid metabolism in hyperbilirubinaemic rats.

## Limitations

In rats, HDL constitutes the greatest lipoprotein fraction involved in plasma cholesterol transport, as opposed to humans where the LDL particle is generally more abundant (Van Der Wulp *et al.* 2013). As such, future studies should evaluate the effects of hyperbilirubinaemia in genetic knockout models that more closely resemble human cholesterol transport such as the LDL knockout mouse model. Nonetheless, increased LDLr expression in female Gunn rats provides preliminary evidence that suggests greater LDL-mediated cholesterol clearance in hyperbilirubinaemia.

## Conclusion

Individuals with elevated circulating concentrations of bilirubin have reduced circulating cholesterol and are protected from CVD. However, very little is under-stood regarding how these affects are mediated *in vivo* (Bulmer *et al.* 2013). This is the first study to demonstrate substantially elevated sterol excretion in female hyperbilirubinaemic Gunn rats. The increased sterol excretion creates a negative intestinal sterol balance that is compensated for by elevated cholesterol synthesis and increased hepatic LDLr expression. Elevated LDLr expression is likely responsible for depleting the circulating cholesterol pool causing hypocholesterolaemia in female Gunn rats. Surprisingly there was significant sexual dimorphism in the changes to sterol metabolism suggesting perturbation of sex hormone metabolism in female UGT1A1-deficient Gunn rats, and therefore, it remains unclear as to whether bilirubin, *per se*, is responsible for these reported hypolipidaemic effects. These data collectively support additional mechanisms to explain cardiovascular disease protection, which could be extrapolated to humans with impaired UGT1A1 function (including GS).

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

## Additional information

### Data availability statement

The raw data supporting the conclusion of this article will be made available by the authors, without undue reservation, to any qualified researcher upon request.

### Competing interests

The authors declare that the research was conducted in the absence of any conflict of interest.

### Author contributions

J.V. designed and performed the research, analysed the data and wrote the manuscript. A.B. assisted with method development, study design and terminal procedures. A.B. also had a primary role in supervision and revision of the manuscript. J.P. assisted with all animal work and biochemistry analysis, and conducted bile duct cannulations, measurement of CYP7A1 and hepatic lipid concentrations. O.R., H.V. and T.D. assisted with planning the study design and method development, and measured the rate of cholesterol synthesis, biliary cholesterol concentrations and biliary bile acid species. A.-C.B. and K.A. conducted PCR analysis. E.P. assisted with the biochemical analysis. J.P., O.R., H.V., T.D., E.P. and K.W. contributed to the interpretation of the results. All authors contributed to the critical revision of the manuscript. All authors approved the final version and submission of this manuscript.

## Acknowledgements

Open access publishing facilitated by Griffith University, as part of the Wiley – Griffith University agreement via the Council of Australian University Librarians.

## Funding

Funding for this research was provided by the School of Pharmacy and Medical Sciences, Griffith University, Gold Coast, Queensland, Australia. This research was also supported by a research grant provided by the Der Wissenschaftsfonds, Austria (FWF, Grant ID: P29608).

## Keywords

bile acid metabolism, cholesterol metabolism, Gilbert's syndrome, LDL receptor, lipid, SREBP, UGT1A1, unconjugated bilirubin

## Supporting information

Additional supporting information can be found online in the Supporting Information section at the end of the HTML view of the article. Supporting information files available:

**Statistical Summary Document**
**Peer Review History**

