## [Peer Review History · The Journal of Physiology]

Sexual Dimorphism: Increased sterol excretion leads to hypocholesterolaemia in female hyperbilirubinaemic Gunn rats

Josif Vidimce, Johara Pillay, Onne Ronda, Ai-Ching Boon, Evan Pennell, Kevin J. Ashton, Theo van Dijk, Karl-Heinz Wagner, Henkjan J. Verkade, and Andrew C. Bulmer

DOI: [10.1113/JP282395](https://doi.org/10.1113/JP282395)

Corresponding author(s): Andrew Bulmer (a.bulmer@griffith.edu.au)

Review Timeline:

Submission Date:	20-Sep-2021
Editorial Decision:	27-Oct-2021
Revision Received:	29-Jan-2022
Accepted:	02-Feb-2022

Senior Editor: Kim Barrett

Reviewing Editor: Kyle McCommis

Transaction Report:

Dear Dr Bulmer,

Re: JP-RP-2021-282395 "Sexual Dimorphism: Increased sterol excretion leads to hypocholesterolaemia in female hyperbilirubinaemic Gunn rats" by Josif Vidimce, Johara Pillay, Onne Ronda, Ai-Ching Boon, Evan Pennell, Kevin J. Ashton, Theo van Dijk, Karl-Heinz Wagner, Henkjan J. Verkade, and Andrew C. Bulmer

Thank you for submitting your manuscript to The Journal of Physiology. It has been assessed by a Reviewing Editor and by 2 expert Referees and I am pleased to tell you that it is considered to be acceptable for publication following satisfactory revision.

The reports are copied at the end of this email. Please address all of the points and incorporate all requested revisions, or explain in your Response to Referees why a change has not been made.

NEW POLICY: In order to improve the transparency of its peer review process The Journal of Physiology publishes online as supporting information the peer review history of all articles accepted for publication. Readers will have access to decision letters, including all Editors' comments and referee reports, for each version of the manuscript and any author responses to peer review comments. Referees can decide whether or not they wish to be named on the peer review history document.

Authors are asked to use The Journal's premium BioRender (<https://biorender.com/>) account to create/redrawn their Abstract Figures. Information on how to access The Journal's premium BioRender account is here: <https://physoc.onlinelibrary.wiley.com/journal/14697793/biorender-access> and authors are expected to use this service. This will enable Authors to download high-resolution versions of their figures.

I hope you will find the comments helpful and have no difficulty returning your revisions within 4 weeks.

Your revised manuscript should be submitted online using the links in Author Tasks Link Not Available.

Any image files uploaded with the previous version are retained on the system. Please ensure you replace or remove all files that have been revised.

REVISION CHECKLIST:

- Article file, including any tables and figure legends, must be in an editable format (eg Word)
- Abstract figure file (see above)
- Statistical Summary Document
- Upload each figure as a separate high quality file
- Upload a full Response to Referees, including a response to any Senior and Reviewing Editor Comments;
- Upload a copy of the manuscript with the changes highlighted.

- A potential 'Cover Art' file for consideration as the Issue's cover image;
- Appropriate Supporting Information (Video, audio or data set https://jp.msubmit.net/cgi-bin/main.plex?form_type=display_requirements#supp).

To create your 'Response to Referees' copy all the reports, including any comments from the Senior and Reviewing Editors, into a Word, or similar, file and respond to each point in colour or CAPITALS and upload this when you submit your revision.

I look forward to receiving your revised submission.

If you have any queries please reply to this email and staff will be happy to assist.

Yours sincerely,

Professor Kim E. Barrett
Editor-in-Chief
The Journal of Physiology
<https://jp.msubmit.net>
<http://jp.physoc.org>
The Physiological Society
Hodgkin Huxley House
30 Farringdon Lane
London, EC1R 3AW
UK
<http://www.physoc.org>
<http://journals.physoc.org>

REQUIRED ITEMS:

-Author photo and profile. First (or joint first) authors are asked to provide a short biography (no more than 100 words for one author or 150 words in total for joint first authors) and a portrait photograph. These should be uploaded and clearly labelled with the revised version of the manuscript. See Information for Authors for further details.

-You must start the Methods section with a paragraph headed Ethical Approval. A detailed explanation of journal policy and regulations on animal experimentation is given in Principles and standards for reporting animal experiments in The Journal of Physiology and Experimental Physiology by David Grundy J Physiol, 593: 2547-2549. doi:10.1113/JP270818.). A checklist outlining these requirements and detailing the information that must be provided in the paper can be found at: <https://physoc.onlinelibrary.wiley.com/hub/animal-experiments>. Authors should confirm in their Methods section that their experiments were carried out according to the guidelines laid down by their institution's animal welfare committee, and conform to the principles and regulations as described in the Editorial by Grundy (2015). The Methods section must contain details of the anaesthetic regime: anaesthetic used, dose and route of administration and method of killing the experimental animals.

-The Reference List must be in Journal format

-Your manuscript must include a complete Additional Information section

-Please upload separate high-quality figure files via the submission form.

-You must upload original, uncropped western blot/gel images (including controls) if they are not included in the manuscript. This is to confirm that no inappropriate, unethical or misleading image manipulation has occurred <https://physoc.onlinelibrary.wiley.com/hub/journal-policies#imagmanip> These should be uploaded as 'Supporting information for review process only'. Please label/highlight the original gels so that we can clearly see which sections/lanes have been used in the manuscript figures.

-Please ensure that any tables are in Word format and are, wherever possible, embedded in the article file itself.

-Please ensure that the Article File you upload is a Word file.

-Your paper contains Supporting Information of a type that we no longer publish. Any information essential to an understanding of the paper must be included as part of the main manuscript and figures. The only Supporting Information that we publish are video and audio, 3D structures, program codes and large data files. Your revised paper will be returned to you if it does not adhere to our Supporting Information Guidelines

-A Statistical Summary Document, summarising the statistics presented in the manuscript, is required upon revision. It must be on the Journal's template, which can be downloaded from the link in the Statistical Summary Document section here: https://jp.msubmit.net/cgi-bin/main.plex?form_type=display_requirements#statistics

-Papers must comply with the Statistics Policy https://jp.msubmit.net/cgi-bin/main.plex?form_type=display_requirements#statistics

In summary:

-If $n \leq 30$, all data points must be plotted in the figure in a way that reveals their range and distribution. A bar graph with data points overlaid, a box and whisker plot or a violin plot (preferably with data points included) are acceptable formats.

-If $n > 30$, then the entire raw dataset must be made available either as supporting information, or hosted on a not-for-profit repository e.g. FigShare, with access details provided in the manuscript.

- n clearly defined (e.g. x cells from y slices in z animals) in the Methods. Authors should be mindful of pseudoreplication.

-All relevant n values must be clearly stated in the main text, figures and tables, and the Statistical Summary Document (required upon revision)

-The most appropriate summary statistic (e.g. mean or median and standard deviation) must be used. Standard Error of the Mean (SEM) alone is not permitted.

-Exact p values must be stated. Authors must not use 'greater than' or 'less than'. Exact p values must be stated to three significant figures even when 'no statistical significance' is claimed.

-Statistics Summary Document completed appropriately upon revision

-Please include an Abstract Figure. The Abstract Figure is a piece of artwork designed to give readers an immediate understanding of the research and should summarise the main conclusions. If possible, the image should be easily 'readable' from left to right or top to bottom. It should show the physiological relevance of the manuscript so readers can assess the importance and content of its findings. Abstract Figures should not merely recapitulate other figures in the manuscript. Please try to keep the diagram as simple as possible and without superfluous information that may distract from the main conclusion(s). Abstract Figures must be provided by authors no later than the revised manuscript stage and should be uploaded as a separate file during online submission labelled as File Type 'Abstract Figure'. Please ensure that you include the figure legend in the main article file. All Abstract Figures should be created using BioRender. Authors should use The Journal's premium BioRender account to export high-resolution images. Details on how to use and access the premium account are included as part of this email.

EDITOR COMMENTS

Reviewing Editor:

Thank you for submitting your manuscript titled "Sexual Dimorphism: Increased sterol excretion leads to hypocholesterolaemia in female hyperbilirubinaemic Gunn rats" to the Journal of Physiology. Your paper has been assessed by two external reviewers, as well as an academic reviewing editor. All reviewers believe this is an important study, investigating an interesting aspect to cholesterol metabolism in male & female rats, as well as across different ages. The external reviewers have requested additional clarifying analyses or experiments, to help better understand these findings. Please also be sure to specify how euthanasia was performed. Also, to conform with Journal statistical policies, please revise tables and figures to provide exact p values instead of asterisks. We hope you find these critiques constructive, and look forward to submission of your revised manuscript.

n is defined on figures and tables, and SD is used. However, please change all figures and tables to provide precise p values instead of asterisks for under 0.05, 0.01, etc.

Senior Editor:

Please see comments from the Reviewing Editor.

REFeree COMMENTS

Referee #1:

Paper by Vidimce et al. entitled "Sexual dimorphism: Increased sterol excretion leads to hypocholesterolaemia in female hyperbilirubinemic Gunn rats" tries to explain relationships between lower serum cholesterol concentrations in hyperbilirubinemic subjects using an animal model of hyperbilirubinemic Gunn rats. The study has been well performed and results are convincing. Nevertheless, there are several important issues which should be addressed prior reconsideration the paper for possible publication in J Physiol.

Major comments

- 1) As authors state, HDL is in rats the major lipoprotein involved in cholesterol transport. However, the authors do not take into account the fact that the fate of cholesterol in the liver is dependent on the source of cholesterol - biliary cholesterol is derived from HDL cholesterol, while bile acids from LDL cholesterol - this is not discussed at all. Instead, the authors argue (p. 33, 2nd para) that greater biliary secretion of cholesterol seen in female Gunn rats could be caused by increased cholesterol synthesis and/or increased coupling of lipids to bile acid secretion, not taking into account the above mentioned fact at all.
- 2) The term "inverse relationship" used several times within the text is incorrect from the mathematical point of view. The inverse relationship is defined as $y = b/x$, whereas a "negative" relationship ($y = a - bx$) is correct for described associations. This should be corrected within the whole text.
- 3) Bilirubin should not be named as a "lipid reducing agent" since there is no clear evidence for such statement. Pls reformulate.
- 4) There is no reason to observe as much as 11 uM concentrations of direct bilirubin in the Gunn rat sera. The most likely explanation is improper method used for its determination. It has been well described that even unconjugated bilirubin might give falsely positive diazo reaction. Thus more accurate method such as HPLC should be used instead. Or better, these results should be entirely removed (they are still not discussed at all).
- 5) Extremely high concentrations of serum bile acids in juvenile rats are confusing -what is the reason? Simultaneously, the reason for very high serum bile acids in the adult female Gunn rats also does not make any sense taking into consideration increased biliary secretion of bile acids as well as the bile flow. The authors need to discuss and explain this discrepancy.
- 6) Large proportion of cholesterol gets into gut lumen via transintestinal cholesterol excretion (TICE). Could authors estimate, based on their data on biliary secretion and total fecal excretion of cholesterol, what is the possible contribution of TICE on fecal cholesterol excretion in Gunn rats?
- 7) Is there any explanation why female Gunn rats should have higher TCA proportion in their biles?
- 8) The authors state that their study "clearly describes functional impact of hyperbilirubinemia/UGT1A1 dysfunction on sterol excretion", they should not ignore the fact that there might be a coinciding genetic defect in the Gunn rats responsible for this observations.

Referee #2:

This manuscript by Vidimce et al explores the mechanisms leading to hypocholesterolemia in Gunn rats, which display hyperbilirubinemia and are a model of Gilbert's syndrome. Specifically, the authors aimed to determine aspects of sterol metabolism in Gunn rats that lead to hypocholesterolemia and to assess whether these were sex-dependent. Analysis of both female and male rats, as juveniles and adults, revealed important interactions of genotype, age, and gender on circulating cholesterol and phospholipids. Further analysis revealed that female Gunn rats had elevations in cholesterol synthesis, biliary lipid secretion, and increased fecal excretion of cholesterol and bile acids. Collectively, the authors explain that increased sterol secretion promotes a negative intestinal sterol balance that leads to increased cholesterol synthesis

and hepatic LDL receptor expression.

These original findings contribute to deciphering important physiological mechanisms surrounding cholesterol metabolism, which could potentially have a great impact on protection from cardiovascular disease. The authors used an appropriate experimental design with appropriate controls; however, a few additional data could greatly strengthen the validity of their conclusions as outlined below.

Comments

-In the methods, it states, "animals expressing hyperbilirubinemia are referred to as "Gunn" rats while littermates with normal bilirubin levels are indicated as "controls". Why were these rats not genotyped? Couldn't this be done for the mutation in Ugt1a1? It would give greater confidence in the validity of these studies if the rats were actually genotyped.

-In the Discussion, the authors state that "Adult female Gunn rats demonstrated a >50% reduction serum cholesterol concentrations compared to controls largely due to a reduction in HDL-C which is the dominant circulating lipoprotein in rodents". It seems that measurements (or calculations) of other major lipoproteins (VLDL and LDL) could be helpful when trying to understand mechanisms leading to hypocholesterolemia. Although not totally necessary for this manuscript, FPLC-separated lipoprotein fractions could help explain why there are lower circulating phospholipids if lipoprotein particle sizes are decreased.

-The Discussion states, "Taken together, these results suggest that female Gunn rats have increased hepatic uptake of circulating cholesterol and elevated cholesterol synthesis, through the enhanced nuclear expression of SREBP2". Except for the 'elevated cholesterol synthesis' this statement is weakly supported by the current data. Hepatic cholesterol uptake was not directly measured. Only LDLr was examined, what about other receptors like SRB1? SREBP2 is a master regulator of genes involved in cholesterol metabolism, but SREBP1a regulates all SREBP-responsive genes (Fig 7) and should also be examined.

-Due to the differences in bile acids and secretion, it seems worthwhile to examine the role of FXR in the female Gunn rats which have increased levels of CDCA conjugates. Are there any variations in FXR targets besides Cyp7a1?

-Authors state that "Bodyweight was recorded every 2 days for the duration of the study", but the data is not shown. Could these data please be added to the supplement?

-Any explanation why the adult female Gunn rats have decreased body weight/food intake, compared to controls?

-It might be helpful to measure hepatic free cholesterol also since there was no change in total cholesterol but an increase in cholesterol synthesis in female Gunn rats.

-In Fig 7D, it wasn't clear why LDLr was normalized to B-actin, while all other targets were normalized to GAPDH.

-Any rationale for differences due to age?

END OF COMMENTS

This manuscript by Vidimce, et al explores the mechanisms leading to hypocholesterolemia in Gunn rats, which display hyperbilirubinemia and are a model of Gilbert's syndrome. Specifically, the authors aimed to determine aspects of sterol metabolism in Gunn rats that lead to hypocholesterolemia and to assess whether these were sex-dependent. Analysis of both female and male rats, as juveniles and adults, revealed important interactions of genotype, age, and gender on circulating cholesterol and phospholipids. Further analysis revealed that female Gunn rats had elevations in cholesterol synthesis, biliary lipid secretion, and increased fecal excretion of cholesterol and bile acids. Collectively, the authors explain that increased sterol secretion promotes a negative intestinal sterol balance that leads to increased cholesterol synthesis and hepatic LDL receptor expression.

These original findings contribute to deciphering important physiological mechanisms surrounding cholesterol metabolism, which could potentially have a great impact in protection from cardiovascular disease. The authors used an appropriate experimental design with appropriate controls; however, a few additional data could greatly strength the validity of their conclusions as outlined below.

Comments

-In the methods, it states, "*animals expressing hyperbilirubinaemia are referred to as "Gunn" rats while littermates with normal bilirubin levels are indicated as "controls"*". Why were these rats not genotyped? Couldn't this be done for the mutation in *Ugt1a1*? It would give greater confidence in the validity of these studies if the rats were actually genotyped.

-In Discussion, authors state that, "Adult female Gunn rats demonstrated a >50% reduction serum cholesterol concentrations compared to controls largely due to a reduction in HDL-C which is the dominant circulating lipoprotein in rodents". It seems that measurements (or calculations) of other major lipoproteins (VLDL and LDL) could be helpful when trying to understand mechanisms leading to hypocholesterolemia. Although not totally necessary for this manuscript, FPLC-separated lipoprotein fractions could help explain why there are lower circulating phospholipids if lipoprotein particle sizes are decreased.

-The Discussion states, "Taken together, these results suggest that female Gunn rats have increased hepatic uptake of circulating cholesterol and elevated cholesterol synthesis, through enhanced nuclear expression of SREBP2". Except for the 'elevated cholesterol synthesis' this statement is weakly supported by the current data. Hepatic cholesterol uptake was not directly measured. Only LDLr was examined, what about other receptors like SRB1? SREBP2 is a master regulator of genes involved in cholesterol metabolism, but SREBP1a regulates all SREBP-responsive genes (Fig 7) and should also be examined.

-Due to the differences in bile acids and secretion, it seems worthwhile to examine the role of FXR in the female Gunn rats which have increased levels of CDCA conjugates. Are there any variations in FXR targets besides Cyp7a1?

-Authors state that "Bodyweight was recorded every 2 days for the duration of the study", but the data is not shown. Could these data please be added to the supplement?

-Any explanation why the adult female Gunn rats have decreased body weight/food intake, compared to controls?

-It might be helpful to measure hepatic free cholesterol also since there was no change in total cholesterol but an increase in cholesterol synthesis in female Gunn rats.

-In Fig 7D, it wasn't clear why LDLr was normalized to B-actin, while all other targets were normalized to GAPDH.

-Any rationale for differences due to age?

Response to Manuscript Editor and Reviewer Comments

Many thanks to the editor and reviewers for their constructive and helpful comments regarding this manuscript. The document below addresses comments in a point-by-point manner, including the original comment, response to comment, and revisions to the manuscript. Changes can be found typed in **orange text** within the manuscript (Cholesterol biosynthesis manuscript - 29012022 - MARKED.docx). A second 'clean' version is also submitted with black text included (Cholesterol biosynthesis manuscript - 29012022 - CLEAN.docx).

Assessor comments in **RED**.

Responses are in **BLUE**.

Original manuscript text is in **BLACK**.

Corrections to the manuscript are in **ORANGE**.

Reviewing Editor:

Thank you for submitting your manuscript titled "Sexual Dimorphism: Increased sterol excretion leads to hypocholesterolaemia in female hyperbilirubinaemic Gunn rats" to the Journal of Physiology. Your paper has been assessed by two external reviewers, as well as an academic reviewing editor. All reviewers believe this is an important study, investigating an interesting aspect to cholesterol metabolism in male & female rats, as well as across different ages. The external reviewers have requested additional clarifying analyses or experiments, to help better understand these findings. Please also be sure to specify how euthanasia was performed. Also, to conform with Journal statistical policies, please revise tables and figures to provide exact p values instead of asterisks. We hope you find these critiques constructive, and look forward to submission of your revised manuscript.

n is defined on figures and tables, and SD is used. However, please change all figures and tables to provide precise p values instead of asterisks for under 0.05, 0.01, etc.

We are pleased to learn that the reviewers found this study important and that it investigated an interesting aspect to cholesterol metabolism. We feel that the comments and suggestions were very useful to help better understand the results. We have adapted the manuscript accordingly. For example, we have revised the methods to include the protocol for euthanasia (see below). We have also modified all figures to provide the precise p values as requested.

Methods (Bile duct cannulation and terminal procedures):

Page 9:

After surgery, blood was collected from the inferior vena cava and centrifuged (2000 g, 10 mins, 4°C). The supernatant was flash frozen in liquid N₂ and stored at -80°C. **Next, the rats were euthanised by surgical removal of the heart.** Liver tissue was rinsed with cold dPBS (Gibco®, United Kingdom) and one section was flash frozen with liquid N₂ and stored at -

80°C while the other was stored in RNAlater solution (Invitrogen, Australia) for at least 24 hrs prior to storage at -80°C.

Reviewer 1:

1) As authors state, HDL is in rats the major lipoprotein involved in cholesterol transport. However, the authors do not take into account the fact that the fate of cholesterol in the liver is dependent on the source of cholesterol - biliary cholesterol is derived from HDL cholesterol, while bile acids from LDL cholesterol - this is not discussed at all. Instead, the authors argue (p. 33, 2nd para) that greater biliary secretion of cholesterol seen in female Gunn rats could be caused by increased cholesterol synthesis and/or increased coupling of lipids to bile acid secretion, not taking into account the above mentioned fact at all.

The origin of biliary cholesterol and of the cholesterol backbone of bile acids has been an area of intensive research for decades, including by some of the authors of the present manuscript. Indeed, there are experimental indications that support preference of HDL-derived cholesterol for biliary cholesterol secretion and a different cholesterol source for biliary bile acid secretion. Although the origins are not clear-cut and sharply defined, we appreciate the comment of the Reviewer and have adapted the Discussion (see *Discussion* on page 33 in revised manuscript).

Discussion:

Page 40:

Intriguingly, adult female Gunn rats also show greater biliary cholesterol secretion. It is tempting to speculate that increased cholesterol synthesis and/or increased HDL-C uptake are involved in this observation (Dijkers & Tietge, 2010). Biliary cholesterol can predominantly originate from HDL-C and the rate of hepatic HDL-C uptake could impact the rate of biliary cholesterol secretion (Dijkers & Tietge, 2010). The profoundly reduced serum HDL-C concentrations in female Gunn rats may be related to reduced HDL production, increased clearance, or both. It seems less likely that female Gunn rats had reduced HDL-C production because hepatic ABCA1 hepatic protein and *Apoa1* gene (Table 4) expression were unchanged. Rather, increased hepatic HDL-C uptake could be related to the increased biliary cholesterol secretion in female Gunn rats (Dijkers & Tietge, 2010). Alternatively, biliary cholesterol secretion may have been increased in female Gunn rats due to increased coupling of lipids to bile acid secretion due to reduced biliary concentration of bilirubin conjugates (Bulmer *et al.*, 2013). Conjugated bilirubin and other organic anions dissociate biliary lipid secretion from bile acids and cause reduction in overall biliary output of cholesterol and phospholipids (Verkade, 2000). However, these studies commonly infused supra-physiological concentrations of organic anions, thus, it is unknown whether increased coupling occurs at physiological or sub-physiological levels of organic anions in bile. The current study presents the first evidence of increased coupling of biliary lipids to bile acid secretion in Gunn rats who secrete substantially lower amounts of bilirubin conjugates in bile. Adult Gunn rats demonstrated greater relative secretion of biliary lipids to bile acids in part explaining the higher biliary cholesterol and phospholipid secretion in females.

2) The term "inverse relationship" used several times within the text is incorrect from the mathematical point of view. The inverse relationship is defined as $y = b/x$, whereas a "negative" relationship ($y = a - bx$) is correct for described associations. This should be corrected within the whole text.

Thank you for the feedback, we have changed "inverse relationship" to "negative relationship" wherever it was used in the manuscript (see page 31).

3) Bilirubin should not be named as a "lipid reducing agent" since there is no clear evidence for such statement. Pls reformulate.

Thank you for the feedback. I believe that the reviewer is referring to the following sentence on page 4: "Recent epidemiological studies show a negative relationship of UCB with total cholesterol and LDL-C, indicating that UCB potentially functions both as an antioxidant and as a lipid reducing agent [12,13]."

We acknowledge that this statement may be mis-interpreted to mean that bilirubin is established as a lipid reducing agent. We have revised this sentence to clarify that the relationship between bilirubin and lipid reduction is mainly associative at this point in time.

Introduction:

Page 5:

Recent epidemiological studies report that UCB is associated with lower total cholesterol and LDL-C, indicating that UCB potentially also regulates cholesterol metabolism in addition to functioning as an antioxidant (Bulmer *et al.*, 2013; Seyed Khoei *et al.*, 2018).

4) There is no reason to observe as much as 11 uM concentrations of direct bilirubin in the Gunn rat sera. The most likely explanation is improper method used for its determination. It has been well described that even unconjugated bilirubin might give falsely positive diazo reaction. Thus more accurate method such as HPLC should be used instead. Or better, these results should be entirely removed (they are still not discussed at all).

Agreed, we have removed the results reporting serum (Table 2) and biliary (Figure 2) direct bilirubin from the manuscript.

5) Extremely high concentrations of serum bile acids in juvenile rats are confusing -what is the reason? Simultaneously, the reason for very high serum bile acids in the adult female Gunn rats also does not make any sense taking into consideration increased biliary secretion of bile acids as well as the bile flow. The authors need to discuss and explain this discrepancy.

Thank you for the feedback, these are important points, please see our responses below.

Why serum bile acid concentrations are higher in juvenile rats:

According to Yousef and Tuchweber (Yousef & Tuchweber, 1982) and Belknap et al. (Belknap *et al.*, 1981) serum bile acid concentrations are higher in juvenile/adolescent rats compared to adult rats with peak concentrations at approximately 22-28 days of age. The reason behind elevated plasma bile acids in juvenile rats is not completely understood but it is potentially caused by a greater total bile acid pool due to a lower capacity of juvenile livers to excrete bile acids (Yousef & Tuchweber, 1982; Morris *et al.*, 1983). Considering that our juvenile rats were between 21-28 days old when bile acid concentrations peak, the relatively higher concentrations in juveniles compared to adult rats is consistent with previous research.

Why serum bile acid concentrations are higher in adult female Gunn rats compared to adult controls:

The higher serum bile acid concentrations in female Gunn rats was an unexpected finding, the reason for which, we can only speculate (the study was not designed to elucidate this). Possibly, UGT1A1 inactivity could elevate circulating oestrogen concentrations (brief summary on this can be found in the discussion [2nd paragraph on page 42]) and thereby downregulating hepatic Na⁺-taurocholate cotransporting polypeptide (NTCP) expression. NTCP is essential for uptake of conjugated bile acids from the plasma that are undergoing enterohepatic circulation (Simon *et al.*, 2004; Li & Dawson, 2020) and its downregulation by oestrogen would lead to elevated serum bile acid concentrations.

6) Large proportion of cholesterol gets into gut lumen via transintestinal cholesterol excretion (TICE). Could authors estimate, based on their data on biliary secretion and total fecal excretion of cholesterol, what is the possible contribution of TICE on fecal cholesterol excretion in Gunn rats?

The Reviewer raises another interesting issue. We would be eager to provide a better estimation of the TICE contribution, but this report does not investigate or report the rate of cholesterol absorption, which would be a necessary parameter to estimate TICE. As described by several of the co-authors (Ronda *et al.*, 2016), TICE is the sum of daily faecal neutral sterol excretion and intestinal cholesterol absorption minus the daily dietary cholesterol intake and biliary cholesterol secretion.

Despite the above limitation, we were able to estimate the net contribution of TICE and cholesterol reabsorption combined. Faecal cholesterol excretion is determined by sum of the biliary cholesterol secretion and TICE subtracted by the rate of cholesterol reabsorption (van de Peppel *et al.*, 2019). Therefore, by subtracting the daily rate of biliary cholesterol secretion from daily faecal cholesterol excretion we can estimate the net contribution of TICE and cholesterol reabsorption and whether this is positive (outward flow) or negative (inward) (see Fig. 4 below). The net contribution of TICE and cholesterol reabsorption can also be defined as the net intestinal cholesterol flux. Estimating the net intestinal cholesterol flux revealed that female Gunn rats have significantly greater (outward) flux into the intestinal lumen, suggesting that cholesterol reabsorption is decreased and/or TICE is increased (see Fig. 4 below). We have incorporated these new results to the manuscript (see below for specific changes) and thank the reviewer for the valuable comment.

Results:

Page 25:

Elevated net intestinal cholesterol flux in female Gunn rats

Faecal cholesterol excretion can be affected by four different mechanisms including 1) dietary cholesterol intake; 2) biliary cholesterol secretion; 3) transintestinal cholesterol secretion (TICE); and 4) cholesterol reabsorption (Fig. 4). In this study, the diet did not contain cholesterol, therefore, faecal cholesterol output was only determined by the latter three mechanisms. Since TICE increases while cholesterol reabsorption reduces the rate of faecal cholesterol excretion, their mutually opposing effects can be defined as a net (outward) intestinal cholesterol flux (TICE – cholesterol reabsorption; Fig. 4) (van de Peppel *et al.*, 2019). Consequently, net intestinal cholesterol flux can be estimated by subtracting biliary cholesterol secretion from faecal cholesterol excretion. There was no effect of phenotype ($p=0.185$) or interaction ($p=0.0931$) on the net intestinal cholesterol flux. However, post-hoc analysis demonstrated that female Gunn rats had significantly elevated net (outward) intestinal cholesterol flux compared to controls ($p=0.0427$; Fig. 4C).

Figure 4. Net intestinal cholesterol flux of adult Gunn (hyperbilirubinaemic) and control (normobilirubinaemic) rats. A) Model describing the four mechanisms (diet, biliary secretion, cholesterol reabsorption, and TICE) that affect the rate of faecal cholesterol excretion. B) The daily rates of biliary cholesterol secretion, net intestinal cholesterol flux, and faecal cholesterol excretion. Since the diet did not contain cholesterol, its contribution was disregarded. The net intestinal cholesterol flux was defined as the overall contribution of TICE and cholesterol reabsorption, and it was estimated by subtracting daily biliary cholesterol secretion from daily faecal cholesterol excretion. C) The daily net intestinal cholesterol flux in a graphical format. Data are presented as means (standard deviation). Two-Way ANOVA was performed with main effects: phenotype (Gunn or control) and sex (male or female). All post-hoc analysis compared differences between phenotypes within the same sex. Statistically significant ($p < 0.05$) p values are highlighted in bold.

Discussion:

Intestinal cholesterol reabsorption is also affected by the rate of bile acid reabsorption. In adult female Gunn rats, a profound increase in faecal bile acid excretion occurred, strongly suggesting reduced intestinal reabsorption of bile acids. ASBT^{-/-} mice with reduced intestinal bile acid reabsorption also demonstrate a substantial decrease in cholesterol reabsorption underpinning the co-dependence of these two pathways (van de Peppel *et al.*, 2019). Indeed, female Gunn rats had elevated net intestinal cholesterol flux which can only be a result of reduced intestinal cholesterol reabsorption and/or by increased TICE (van de Peppel *et al.*, 2019). Consequently, it is tempting to speculate that cholesterol reabsorption is reduced in female Gunn rats, however, it is not possible to separate the individual contributions of the two mechanisms. Collectively, the findings of this study support a conclusion that perturbation in cholesterol balance in female Gunn rats is caused by increased excretion of bile acids and to a lesser extent of cholesterol. The resulting negative intestinal sterol balance is compensated for by elevated cholesterol synthesis.

7) Is there any explanation why female Gunn rats should have higher TCA proportion in their biles?

We assume that the Reviewer refers to the *lower* taurocholic acid (TCA) in bile of female Gunn rats. At this stage the answer to this question remains unknown, and we have acknowledged this in the new version of the manuscript (see below changes to manuscript Discussion). However, we can speculate as to why this may have occurred, please see our explanation below.

Considering that female Gunn rats had significantly higher faecal bile acid excretion, which could not be completely explained by their elevated biliary bile acid secretion, we speculate that intestinal bile acid reabsorption was reduced. Approximately 95% of secreted bile acids are reabsorbed by passive diffusion or by active transport through the apical sodium-dependent bile acid transporter (ASBT) in the distal ileum (Hofmann, 2009). Active transport is the predominant mechanism of bile acid reabsorption. Therefore, we speculate that there was reduced expression/function of ASBT in female Gunn rats that led to increased faecal bile acid excretion. As a result of reduced ASBT-mediated reabsorption, the bile acid species reabsorbed through passive diffusion would have a greater contribution to the biliary bile acid pool than normal. De-conjugated chenodeoxycholic acid (CDCA), lithocholic acid (LCA), and deoxycholic acid (DCA) bile acid species more readily passively diffuse across monolayers compared to cholic acid (CA) *in vitro* (Balakrishnan *et al.*, 2006). In support of this we observed a greater proportion of CDCA species in the bile in female Gunn rats (see Figure 5 on page 24 in the manuscript). Similarly, ASBT knockout mice demonstrate a reduced proportion of TCA with an increase in DCA in their bile compared to wild-type mice (van de Peppel *et al.*, 2019).

Discussion:

Page 42:

In comparison to hydrophilic bile acids (e.g. α/β -MCA species), hydrophobic species such as CA and TCA are potent inducers of cholesterol absorption (Wang et al., 2003). Profoundly lower proportion of TCA was measured in the bile of female Gunn rats (minor elevation in MCA species). The reason for the difference in the proportions of bile acid species in female Gunn rats remains unknown, however, the overall hydrophobicity index did not change. These data suggest that the difference in biliary bile acid composition in female Gunn rats, unlikely affects cholesterol reabsorption.

8) The authors state that their study "clearly describes functional impact of hyperbilirubinemia/UGT1A1 dysfunction on sterol excretion", they should not ignore the fact that there might be a coinciding genetic defect in the Gunn rats responsible for this observations.

Thank you for the important comment. However, we regularly back cross our animals to genetic wild types (Wistar rats) minimising the likelihood of coinciding genetic defects in our animals. Furthermore, the biochemical measurements (e.g. circulating cholesterol concentrations) in this study agree with other published studies with independent colony sources of Gunn rats i.e. investigators from Czech Republic (Wallner *et al.*, 2013).

Reviewer 2:

1. In the methods, it states, "animals expressing hyperbilirubinemia are referred to as "Gunn" rats while littermates with normal bilirubin levels are indicated as "controls". Why were these rats not genotyped? Couldn't this be done for the mutation in *Ugt1a1*? It would give greater confidence in the validity of these studies if the rats were actually genotyped.

Thank you for the comment. As an example of genotyping for UGT1A1 in this rat colony is provided below. Hyperbilirubinaemia was consistently induced in homozygote Gunn rat, therefore, it was not necessary to genotype all animals.

Please see figure below of genotyping evidence of 8 animals used in this study (4 heterozygote normal bilirubin rats [lanes 4-7] and 4 homozygote hyperbilirubinaemic Gunn rats [lanes 8-11]).

Figure 1. Agarose gel electrophoretic analysis of products derived from PCR amplification. BstNI digested samples contain one copy of the wild-type sequence, producing fragments of 82 and 227 bp. The undigested product at 309 bp demonstrates the presence of mutant alleles. The product sizes are: homozygous Gunn, 309 bp; homozygous wild-type Wistar (not shown here), 227 and 82 bp only; heterozygous Wistar 309, 227 and 82 bp. Lane 1: 25bp ladder, lane 2: blank, lane 3: No template control, lanes 4-7: heterozygous Wistar, lanes 8-11: homozygous Gunn.

We have modified the Methods in the manuscript to indicate this statement (please see below).

Methods:

Page 7:

Gunn rats were determined phenotypically to be homozygote based on the presence of jaundice in the days after birth and were ear-tagged. The hyperbilirubinaemic phenotype was then confirmed by measuring serum bilirubin concentrations. **This Gunn colony is regularly backcrossed with female wild-type Wistar rats, supplied by the Animal Resources Centre (Canning Vale, WA, Australia).** From this point, animals expressing hyperbilirubinaemia are referred to as “Gunn” rats while littermates with normal bilirubin levels are indicated as “controls”.

2. In the Discussion, the authors state that "Adult female Gunn rats demonstrated a >50%

reduction serum cholesterol concentrations compared to controls largely due to a reduction in HDL-C which is the dominant circulating lipoprotein in rodents". It seems that measurements (or calculations) of other major lipoproteins (VLDL and LDL) could be helpful when trying to understand mechanisms leading to hypocholesterolemia. Although not totally necessary for this manuscript, FPLC-separated lipoprotein fractions could help explain why there are lower circulating phospholipids if lipoprotein particle sizes are decreased.

Thank you for the suggestion, unfortunately we did not have sufficient aliquots to perform such measurements, although we appreciate the value that this data would bring.

3. The Discussion states, "Taken together, these results suggest that female Gunn rats have increased hepatic uptake of circulating cholesterol and elevated cholesterol synthesis, through the enhanced nuclear expression of SREBP2". Except for the 'elevated cholesterol synthesis' this statement is weakly supported by the current data. Hepatic cholesterol uptake was not directly measured. Only LDLr was examined, what about other receptors like SRB1? SREBP2 is a master regulator of genes involved in cholesterol metabolism, but SREBP1a regulates all SREBP-responsive genes (Fig 7) and should also be examined.

We agree with the Reviewer about the speculative nature in the statement quoted by the reviewer. Accordingly, we have revised the statement in our discussion (please see Page 34 in manuscript and directly below) to better reflect the speculative tone.

Discussion:

Page 42:

Taken together, these results suggest that female Gunn rats have increased hepatic uptake of circulating cholesterol and elevated cholesterol synthesis, and this could be mediated through enhanced nuclear expression of SREBP2. However, to confirm this, future studies need to investigate the expression of other important cholesterol uptake receptors such as scavenger receptor class B type 1 and transcription factors such as SREBP1a. Nevertheless, this pattern of gene/protein regulation is a typical physiological response to reduced hepatic cholesterol content that aims to restore intracellular cholesterol as reported with lipid-lowering therapies (Schonewille *et al.*, 2016).

4. Due to the differences in bile acids and secretion, it seems worthwhile to examine the role of FXR in the female Gunn rats which have increased levels of CDCA conjugates. Are there any variations in FXR targets besides Cyp7a1?

Thank you for the comment. The hepatic gene expression of FXR itself was not different between female animals (please see *NR1H4* expression in Table 4 in manuscript), and downstream targets (Wang *et al.*, 2008) *Cyp7a1*, *Apoa1*, *ApoE*, *Vldlr*, *Insig2*, and *Apoc3* (please see gene expression results in Figure 7 and Table 4 of manuscript) were also not significantly different suggesting that FXR function was not affected in these animals.

5. Authors state that "Bodyweight was recorded every 2 days for the duration of the study", but the data is not shown. Could these data please be added to the supplement?

Thank you for the comment. The figure representing bodyweight for the duration of the study has been added to the supplementary document (see Figure S1 in Supporting information for review process only).

6. Any explanation why the adult female Gunn rats have decreased body weight/food intake, compared to controls?

There is no explanation for why the animals consume less food, however, female Gunn rats report increased fatty acid oxidation which may explain the reduced bodyweight gain over time (please see (Vidimce *et al.*, 2021)).

7. It might be helpful to measure hepatic free cholesterol also since there was no change in total cholesterol but an increase in cholesterol synthesis in female Gunn rats.

Thank you for the comment, hepatic tissues are no longer available, however, we will aim to complete this in future studies.

8. In Fig 7D, it wasn't clear why LDLr was normalized to B-actin, while all other targets were normalized to GAPDH.

Thank you for the comment. For technical reasons, the range of linearity (i.e. the range of linear rise in fluorescent signal over increasing protein concentrations) of B-actin was only suitable for LDLr measurement, however, it was not suitable for the protein concentrations we needed to use for protein targets HMGCR, CYB5R3, CYP7A1, SREBP2, and ABCA1. In contrast, GAPDH had a wider linearity range which enabled its use for the quantification of the aforementioned target proteins.

9. Any rationale for differences due to age?

We agree with the reviewer that the age of Gunn rats in relation to the physiological effects are worthy of further investigation. Please see also our response to question 5 of Reviewer #1 regarding circulating bile acid concentrations.

Many thanks for the constructive comments and opportunity to revise this manuscript and we look forward to receiving the editor's response in due course.

Yours Sincerely,

Andrew C. Bulmer, *PhD* and Josif Vidimce, *PhD*.

References

- Balakrishnan A, Wring SA & Polli JE (2006). Interaction of native bile acids with human apical sodium-dependent bile acid transporter (hASBT): Influence of steroidal hydroxylation pattern and C-24 conjugation. *Pharm Res* **23**, 1451–1459.
- Belknap WM, Balistreri WF, Suchy FJ & Miller PC (1981). Physiologic cholestasis II: Serum bile acid levels reflect the development of the enterohepatic circulation in rats.

Hepatology **1**, 613–616.

- Dijkers A & Tietge UJF (2010). Biliary cholesterol secretion: More than a simple ABC. *World J Gastroenterol* **16**, 5936–5945.
- Hofmann AF (2009). The enterohepatic circulation of bile acids in mammals: form and functions. *Front Biosci (Landmark Ed)* **14**, 2584–2598.
- Li J & Dawson PA (2020). Animal Models to Study Bile Acid Metabolism. *Biochim Biophys Acta - Mol Basis Dis* **1865**, 895–911.
- Morris AI, Little JM & Lester R (1983). Development of the Bile Acid Pool in Rats from Neonatal Life through Puberty to Maturity. *Digestion* **28**, 216–224.
- van de Peppel IP, Bertolini A, van Dijk TH, Groen AK, Jonker JW & Verkade HJ (2019). Efficient reabsorption of transintestinally excreted cholesterol is a strong determinant for cholesterol disposal in mice. *J Lipid Res* **60**, 1562–1572.
- Ronda OAH, van Dijk TH, Verkade HJ & Groen AK (2016). Measurement of Intestinal and Peripheral Cholesterol Fluxes by a Dual-Tracer Balance Method. *Curr Protoc Mouse Biol* **6**, 408–434.
- Simon FR, Fortune J, Iwahashi M, Qadri I & Sutherland E (2004). Multihormonal regulation of hepatic sinusoidal Ntcp gene expression. *Am J Physiol Liver Physiol* **287**, G782–G794.
- Vidimce J, Pillay J, Shrestha N, Dong L, Neuzil J, Wagner K-H, Holland OJ & Bulmer AC (2021). Mitochondrial Function, Fatty Acid Metabolism, and Body Composition in the Hyperbilirubinemic Gunn Rat. *Front Pharmacol* **12**, 1–18.
- Wallner M, Marculescu R, Doberer D, Wolzt M, Wagner O, Vitek L, Bulmer AC & Wagner KH (2013). Protection from age-related increase in lipid biomarkers and inflammation contributes to cardiovascular protection in Gilbert's syndrome. *Clin Sci* **125**, 257–264.
- Wang Y-D, Chen W-D, Moore DD & Huang W (2008). FXR: a metabolic regulator and cell protector. *Cell Res* **18**, 1087–1095.
- Yousef IM & Tuchweber B (1982). Bile Acid Composition in Neonatal Life in Rats. *Neonatology* **42**, 105–112.

Dear Dr Bulmer,

Re: JP-RP-2022-282395R1 "Sexual Dimorphism: Increased sterol excretion leads to hypocholesterolaemia in female hyperbilirubinaemic Gunn rats" by Josif Vidimce, Johara Pillay, Onne Ronda, Ai-Ching Boon, Evan Pennell, Kevin J. Ashton, Theo van Dijk, Karl-Heinz Wagner, Henkjan J. Verkade, and Andrew C. Bulmer

I am pleased to tell you that your paper has been accepted for publication in The Journal of Physiology.

NEW POLICY: In order to improve the transparency of its peer review process The Journal of Physiology publishes online as supporting information the peer review history of all articles accepted for publication. Readers will have access to decision letters, including all Editors' comments and referee reports, for each version of the manuscript and any author responses to peer review comments. Referees can decide whether or not they wish to be named on the peer review history document.

Are you on Twitter? Once your paper is online, why not share your achievement with your followers. Please tag The Journal (@jphysiol) in any tweets and we will share your accepted paper with our 23,000+ followers!

The last Word version of the paper submitted will be used by the Production Editors to prepare your proof. When this is ready you will receive an email containing a link to Wiley's Online Proofing System. The proof should be checked and corrected as quickly as possible.

Authors should note that it is too late at this point to offer corrections prior to proofing. The accepted version will be published online, ahead of the copy edited and typeset version being made available. Major corrections at proof stage, such as changes to figures, will be referred to the Reviewing Editor for approval before they can be incorporated. Only minor changes, such as to style and consistency, should be made a proof stage. Changes that need to be made after proof stage will usually require a formal correction notice.

All queries at proof stage should be sent to TJP@wiley.com

Yours sincerely,

Professor Kim E. Barrett
Editor-in-Chief
The Journal of Physiology
<https://jp.msubmit.net>
<http://jp.physoc.org>
The Physiological Society
Hodgkin Huxley House
30 Farringdon Lane
London, EC1R 3AW
UK
<http://www.physoc.org>
<http://journals.physoc.org>

P.S. - You can help your research get the attention it deserves! Check out Wiley's free Promotion Guide for best-practice recommendations for promoting your work at www.wileyauthors.com/eoo/guide. And learn more about Wiley Editing Services which offers professional video, design, and writing services to create shareable video abstracts, infographics, conference posters, lay summaries, and research news stories for your research at www.wileyauthors.com/eoo/promotion.

*** IMPORTANT NOTICE ABOUT OPEN ACCESS ***

Information about Open Access policies can be found here <https://physoc.onlinelibrary.wiley.com/hub/access-policies>

To assist authors whose funding agencies mandate public access to published research findings sooner than 12 months after publication The Journal of Physiology allows authors to pay an open access (OA) fee to have their papers made freely available immediately on publication.

You will receive an email from Wiley with details on how to register or log-in to Wiley Authors Services where you will be able to place an OnlineOpen order.

You can check if your funder or institution has a Wiley Open Access Account here <https://authorservices.wiley.com/author-resources/Journal-Authors/licensing-and-open-access/open-access/author-compliance-tool.html>

Your article will be made Open Access upon publication, or as soon as payment is received.

If you wish to put your paper on an OA website such as PMC or UKPMC or your institutional repository within 12 months of publication you must pay the open access fee, which covers the cost of publication.

OnlineOpen articles are deposited in PubMed Central (PMC) and PMC mirror sites. Authors of OnlineOpen articles are permitted to post the final, published PDF of their article on a website, institutional repository, or other free public server, immediately on publication.

Note to NIH-funded authors: The Journal of Physiology is published on PMC 12 months after publication, NIH-funded authors DO NOT NEED to pay to publish and DO NOT NEED to post their accepted papers on PMC.

EDITOR COMMENTS

Reviewing Editor:

Thank you for your resubmission of your manuscript titled "Sexual Dimorphism: Increased sterol excretion leads to hypocholesterolaemia in female hyperbilirubinaemic Gunn rats" to The Journal of Physiology. Both academic reviewers and the reviewing editor are satisfied with the responses to previous critiques and the revised manuscript. And thank you for the revisions to comply with The Journal's statistical policy. We are now happy to accept this manuscript for publication.

REFEREE COMMENTS

Referee #1:

The Authors addressed all the issues, I do not have any further questions.

Referee #2:

The authors have satisfied all my previous concerns. Great job.

1st Confidential Review

29-Jan-2022